# Computational stabilization of T cell receptors allows pairing with antibodies to form bispecifics

Karen Froning[1,4], Jack Maguire[2,4], Arlene Sereno[1], Flora Huang[1], Shawn Chang[1], Kenneth Weichert[1], Anton J. Frommelt[1], Jessica Dong[1], Xiufeng Wu[1], Heather Austin[1], Elaine M. Conner[1], Jonathan R. Fitchett[1], Aik Roy Heng[1], Deepa Balasubramaniam[1], Mark T. Hilgers[1], Brian Kuhlman[3✉] & Stephen J. Demarest[1✉]

Recombinant T cell receptors (TCRs) can be used to redirect naïve T cells to eliminate virally infected or cancerous cells; however, they are plagued by low stability and uneven expression. Here, we use molecular modeling to identify mutations in the TCR constant domains (Cα/Cβ) that increase the unfolding temperature of Cα/Cβ by 20 °C, improve the expression of four separate α/β TCRs by 3- to 10-fold, and improve the assembly and stability of TCRs with poor intrinsic stability. The stabilizing mutations rescue the expression of TCRs destabilized through variable domain mutation. The improved stability and folding of the TCRs reduces glycosylation, perhaps through conformational stabilization that restricts access to N-linked glycosylation enzymes. The Cα/Cβ mutations enables antibody-like expression and assembly of well-behaved bispecific molecules that combine an anti-CD3 antibody with the stabilized TCR. These TCR/CD3 bispecifics can redirect T cells to kill tumor cells with target HLA/peptide on their surfaces in vitro.

---

[1] Eli Lilly Biotechnology Center, 10300 Campus Point Drive, San Diego, CA 92121, USA. [2] Program in Bioinformatics and Computational Biology, University of North Carolina at Chapel Hill, Chapel Hill, North Carolina, USA. [3] Department of Biochemistry and Biophysics and Lineberger Comprehensive Cancer Center, University of North Carolina at Chapel Hill, Chapel Hill, NC, USA. [4]These authors contributed equally: Karen Froning, Jack Maguire. ✉email: bkuhlman@email.unc.edu; demarestsj@lilly.com

T cell receptors (TCRs) are the adaptive immune system's tool for recognizing and eliminating "non-self" intracellular antigens. These non-self antigens can emerge from viral infection or from genetic alteration. Genetic alterations are key to aberrant cellular function and can lead to diseases such as cancer. α/β TCRs are expressed on both CD8+ effector and CD4+ helper T cells and recognize proteosomally degraded foreign antigens displayed on infected/cancerous cell surfaces when complexed with HLA/MHC[1,2]. When expressed on CD8+ T effector cells, α/β TCRs interact specifically with Type I MHC/peptide complexes and this interaction induces T cell activation and elimination of cells displaying recognizable non-self antigens.

Many approaches have been developed to harness the exquisite non-self-recognizing properties of α/β TCRs for therapeutic use. Recombinant α/β TCRs have been transduced/transfected into bulk naïve T cells as a means of retargeting these T cells to target tumor associated antigens[3,4]. Alternately, the use of the soluble extracellular region of the TCR fused to an scFv that recognizes an activating TCR subunit, commonly CD3ε, has been used to redirect endogenous T cells to target tumor cells[5]. Unlike antibody-based Bispecific T cell Engagers (BiTEs) that rely on the direct recognition of overexpressed antigens on the cell surface[6], TCR or TCR-mimic bispecifics can recognize a much larger subset of intracellular and abnormal tumor or viral antigens and thus have broader potential applicability[5].

However, TCR assembly and expression is challenging[7]. The common method of soluble α/β TCR production is expression as inclusion bodies in Escherichia coli (E. coli) followed by resolubilization, refolding/assembly/oxidation, and finally purification at relatively low yields[8]. To simplify the assembly, some researchers have tried using only the variable domain regions of TCRs in a single chain format or scTv, like an antibody single chain Fv (scFv), for targeting specific HLA/peptide complexes[9]. While scFvs have shown a propensity for instability, aggregation, and low solubility, scTvs have generally shown even worse expression and stability issues. Great efforts have been made to stabilize scTvs for therapeutic and diagnostic use[9]. Unfortunately, the high diversity of Vα/Vβ germlines (higher than antibody $V_H$/$V_L$ germlines), renders each set of stabilizing designs within scTvs to be unique to the individual Vα/Vβ subunit and unlikely to find general use.

Given that most extracellular proteins are intrinsically glycosylated with complex disulfide pairings, we decided to rely on mammalian expression to generate soluble TCRs. Industrial antibody production has predominately moved to mammalian expression systems[10]. However, it has been our experience that α/β TCRs express poorly with less reliable assembly than antibodies when expressed in the commonly utilized Chinese hamster ovary (CHO) cell system. Many novel bispecific antibody (BsAb) formats, including those with relevance to soluble α/β TCR bispecifics, may require TCRs to express at antibody-like levels for proper molecular assembly. Bispecific "ImmTac" moieties that recombinantly fuse soluble TCRs to antibody scFvs have typically followed the complex E. coli production process described above[5].

We hypothesize that general stabilization of the Cα/Cβ subunit may improve the overall stability and folding of α/β TCRs. Recent studies have shown that strong thermodynamic cooperativity exists between the subunits of α/β TCRs. Cα requires pairing with Cβ in the ER for folding similar to what has been observed for antibody $C_H1$/Cκ subunits[11,12]. Additionally, many Vα/Vβ subunits are intrinsically unfolded in isolation and require Cα/Cβ for proper folding[13]. In support of our hypothesis that Cα/Cβ stabilization may generally improve TCR expression and stability, adding a disulfide between the Cα/Cβ domains positively impacts many α/β TCRs[14]. Therefore, we set out to design a robust Cα/Cβ subunit for general TCR stabilization with the goal of generating TCRs at antibody-like levels that assemble properly.

To identify mutations that stabilize the Cα and Cβ domains, we perform protein design simulations with the molecular modeling software Rosetta[15]. During a design simulation, Rosetta samples alternative amino acid sequences and sidechain conformations in search of mutations that lower the calculated energy of the protein[16]. The Rosetta energy function favors amino acids that pack well and form favorable electrostatic and hydrogen bonding interactions while minimizing desolvation costs and torsional strain[17]. It is common to experimentally test several predictions in search of the best performing mutations because it is difficult to accurately predict how a mutation will affect protein stability[18]. An alternative strategy for finding mutations that will stabilize a protein is to assemble a multiple sequence alignment (MSA) for the protein family and search for highly conserved amino acids that are not conserved in the protein of interest[19]. Recently, there has been considerable success in finding stabilizing mutations by combining conservation analysis with energy-based methods like Rosetta[20]. One potential advantage of the MSA-based approach is that it may capture phenomena, such as the role of a residue in preventing aggregation, that are difficult to capture with a structure-based approach. Here, we test mutations based solely on Rosetta calculations as well as use conservation analysis to filter the results from the design simulations. After screening a host of computationally designed mutations in the Cα/Cβ context, we identify seven mutations that, when combined, significantly improve Cα/Cβ and full-length α/β TCR assembly and expression. These stabilized TCRs can be fused to antibody domains to generate functional BsAbs.

## Results

**Stabilizing the Cα/Cβ TCR subunit.** First, we investigated the thermodynamic properties of a soluble TCR, 1G4_122, and its Vα/Vβ and Cα/Cβ subunits. 1G4_122 binds the NY-ESO-1 antigen[21]. Using a mammalian expression system, we generated both the Vα/Vβ and Cα/Cβ subunits in the presence and absence of flexible (Gly4Ser×4) linkers that link Vα to Vβ or Cα to Cβ. We also tested the subunit expression and assembly with or without stabilizing interdomain disulfides. Most of the Vα/Vβ and Cα/Cβ constructs we generated either failed to express or failed to assemble, including the single chain variants. The best Vα/Vβ subunit expression was obtained by adding a Vα44/Vβ110 disulfide (homologous to the $V_H44$/$V_L100$ disulfide used to stabilize antibody variable domain fragments or Fvs[22]), while the best Cα/Cβ expression was obtained using both the known stabilizing disulfide (Cα166/Cβ173) near the top of the Cα/Cβ interface and the native Cα/Cβ disulfide at the C-terminus of the domains (Cα213/Cβ247)[14]. Differential scanning calorimetry (DSC) experiments showed the (Cα166/Cβ173) disulfide increased the midpoint of thermal unfolding ($T_m$) of the full-length TCR (both subunits) by 8 °C. When comparing the Vα/Vβ and Cα/Cβ subunit unfolding to the unfolding of the full-length extracellular TCR protein, the individual subunits were clearly both destabilized in the absence of one another (Fig. 1a), agreeing with the thermodynamic cooperativity observed previously[13].

To investigate if further stabilization of Cα/Cβ could benefit TCR expression, stability, and assembly when expressing soluble TCRs in mammalian cells, we used molecular modeling simulations to identify mutations that stabilize the constant domains. We scanned over every residue position in the constant domains and used Rosetta to model all possible point mutations (except for cysteine). For each simulation, the TCR was fixed in space and only residues in close proximity to the mutation were allowed to make small movements ("relax") to accommodate the mutation. Long-range changes in structure were not allowed to reduce noise in the energy calculations. We repeated these local relaxations

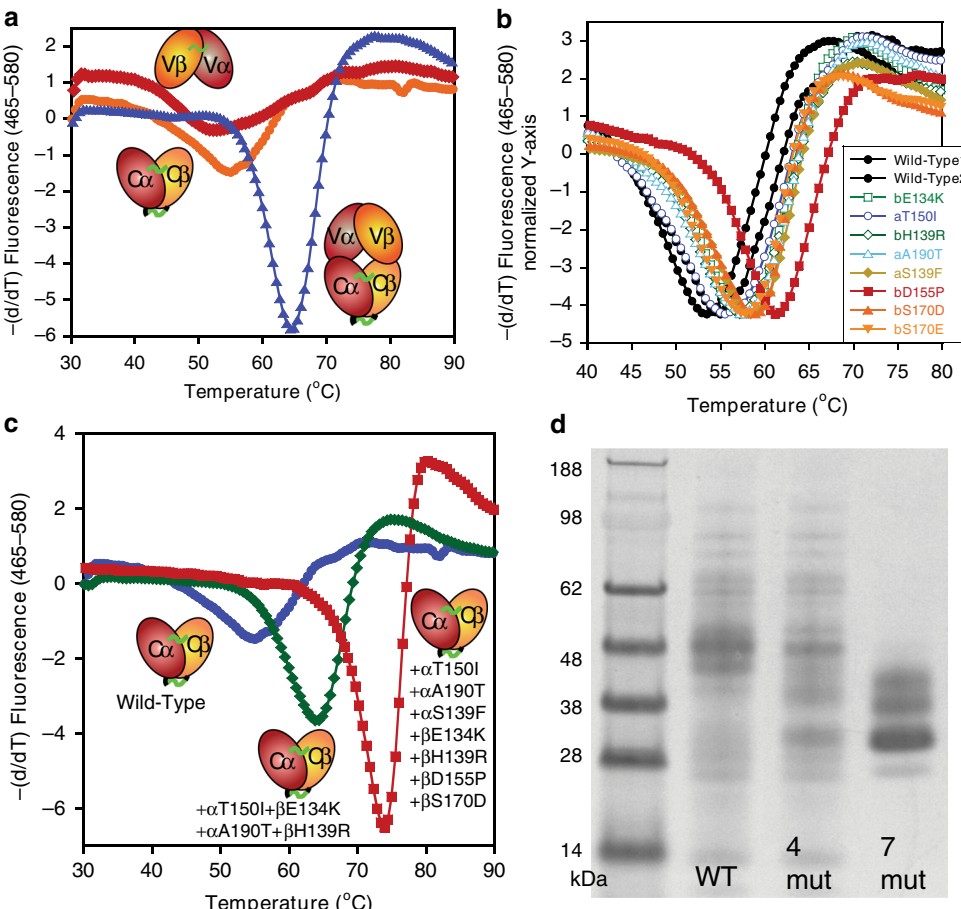

**Fig. 1 Characterization of TCR fragments and stabilizing designs. a** DSF denaturation curves with recombinant TCR (1G4_122, ▲), Vα/Vβ fragment from the same TCR (♦) with an α42/β110 engineered disulfide (homologous to the VH44/VL100 disulfide in dsFvs), and Cα/Cβ fragment with both the standard stalk disulfide and a stabilizing α166/β173 disulfide (●). **b** DSF denaturation curves of the "wild-type" (WT) Cα/Cβ subunit and Cα/Cβ subunits containing individual stabilizing mutations. **c** DSF curves of wild-type Cα/Cβ (●) and Cα/Cβ with 4 (♦) and 7 (■) stabilizing mutations. All points on the DSF plots are the single measured fluorescence values at each temperature. **d** SDS-PAGE characterization of wild-type Cα/Cβ (left lane) and Cα/Cβ with four (middle lane) and seven (right lane) stabilizing mutations. While this direct comparison by SDS-PAGE was performed once, at least two preparations (>3 for the wild-type Cα/Cβ) were produced for each protein yielding similar SDS-PAGE results. Source data are available in the Source Data file.

using the native sequence without any mutations. The predicted change in energy was determined by comparing the Rosetta score of the mutation with the Rosetta score of the native sequence. We then split the mutations into two lists: a list of mutations that are observed in a MSA of TCRs and a list of mutations that are absent from the MSA. We then picked the mutations with the best predicted energies from both lists for experimental screening. This method allowed a conserved mutation to be tested even if it did not have the best Rosetta energy.

Using the isolated Cα/Cβ fragment containing the Cα166/Cβ173 disulfide bond (Fig. 1a), the library of single, computationally designed mutants was generated and expressed in two separate blocks. Increased expression level was used blindly to select mutants for scale-up, purification, and characterization by differential scanning fluorimetry (DSF, Supplementary Table S1). Expression was assessed using an Octet Red with Ni$^{2+}$-NTA sensortips and purified Cα/Cβ protein containing a N-terminal Cα-histag for the standard curve. Increased expression correlated well with increased protein stability. Roughly half of the single mutants chosen for further evaluation based on expression demonstrated a significant increase in stability over the wild-type (native) Cα/Cβ protein (Supplementary Table S1). The $T_m$s for seven of the identified stabilizing mutants are shown in Fig. 1b and range from +1 to +7 °C over wild-type Cα/Cβ. Five of these

mutations were substitutions that are observed in a MSA of TCRs, but the two mutations that produced the biggest gains in $T_m$, D155P (+7.1 °C) and S139F (+4.4 °C), are not observed in the MSA. Supplementary Fig. S1 shows the experimental outcome of each modeled point mutation with respect to its computational metrics.

Next, the stabilizing mutations were combined into a variant harboring four of the mutations and a variant with all seven mutations. Modeling results suggested each of the mutations was likely compatible with the other mutations (i.e., unlikely to interfere with each other). Adding all seven stabilizing mutations led to a seven-fold increase in expression and 20 °C increase in the $T_m$ over the Cα/Cβ subunit harboring only the Cα166/Cβ173 disulfide (denoted "wild-type", Table 1, Fig. 1c). Assembly of the wild-type and variant Cα/Cβ subunits was probed on a non-reducing SDS-PAGE gel. The Cα/Cβ variant with seven mutations assembled more efficiently as evidenced by the lack of disulfide laddering and was smaller/more compact, likely due to less spurious glycosylation (described in detail below) via maintenance of a more discretely folded and compact structure (Fig. 1d). Even for the stabilized variant, multiple bands are clear on the SDS-PAGE gel (Fig. 1d) that likely reflect partial glycan occupancy of the three N-linked glycosylation sites in Cα and single N-linked glycosylation site in Cβ.

**Table 1 Expression and thermal stability of wild-type (WT) TCRs and TCRs with the seven stabilizing mutations (Des).**

| | Titer (mg/L) | Titer ratio[a] (Variant/WT) | DSF or DSC $T_m$ (°C) |
|---|---|---|---|
| Cα/Cβ Subunit with the αT166C/βS173C disulfide | | | |
| WT | 7.3 | 1 | 54.4 ± 1.0 |
| Four mutants | 12.1 ± 8 | 1.66 | 64.1 |
| Seven mutants (Des) | 54 ± 6 | 7.3 | 73.4 |
| Full-Length TCRs without the αT166C/βS173C disulfide | | | |
| NY-ESO-1 WT | 9.5 | 1 | 53.0 |
| NY-ESO-1 Des | 104 | 10.9 | 58.6, 63.8 |
| MAGE A3 WT | 71 ± 9 | 1 | 54.7 |
| MAGE A3 Des | 190 ± 65 | 2.7 | 63.2 |
| HIV T36-5 WT | 35 ± 8 | 1 | Poor assembly |
| HIV T36-5 Des | 232 ± 110 | 6.6 | 58.6, 61.6 |
| WT-1 CE10 WT | 72 ± 39 | 1 | Poor assembly |
| WT-1 CE10 Des | 430 ± 168 | 6.0 | 58.8, 65.4 |
| Full-Length TCRs with the αT166C/βS173C disulfide[b] | | | |
| NY-ESO-1 WT | 24 | 1 | 60.8, 63.7 |
| NY-ESO-1 Des | 85 | 3.5 | 59.5, 65.2 |
| MAGE_A3 WT | 29 | 1 | 61.5 |
| MAGE_A3 Des | 80 | 2.76 | 61.4, 66.5 |
| HIV_T36-5 WT | 30 | 1 | 53.5 |
| HIV_T36-5 Des | 80 | 2.67 | 57.3, 67.6 |
| WT-1_CE10 WT | 25 | 1 | 55.9, 61.1 |
| WT-1_CE10 Des | 70 | 2.8 | 60.6, 69.0 |

[a]If averages±errors are provided, these are the mean of two expressions and the error is the difference between the two values.
[b]The TCRs containing the αT166C/βS173C disulfide were transfected in a single experiment, but not at the same time as the TCRs without the disulfide. Due to subtle variances in CHO cell viability, CHO cell passage, media, and other factors, comparison of expressed titers from experiment to experiment can be misleading, but intra-experimental comparisons are valid.

Lastly, we assessed the impact of the designs on potential immunogenicity of the TCR. Using the EpiVax server, we investigated whether any increase in MHC-peptide complexes were likely for each of the mutations[23]. Overall, a slight increase in the EpiMatrix score was observed for both Cα and Cβ. Increases in the individual peptide scores were marginal and did not reach the level predicted to result in significant MHC binding. An outlier of concern included the Cα_T150I mutation, whose score went up substantially for one peptide 9-mer and there is strong predicted MHC binding predicted for a second 9-mer peptide that also exists in the wild-type Cα peptide. The Cβ_E134K_H139R mutants are found together on multiple 9-mer peptides and there is a slight increase in the EpiMatrix scores, but none of the scores put these peptides in the high-risk category (i.e., strong predicted MHC binding). Interestingly, the Cβ_D155P and Cβ_S170D variants, which are the two most impactful mutants from a $T_m$ perspective, lower the overall EpiMatrix scores versus wild-type Cα/Cβ.

**Characterization of the seven stabilizing Cα/Cβ mutations.** Next, we attempted to crystallize both the wild-type and stabilized forms of the Cα/Cβ subunit. N-linked glycans were removed enzymatically before crystallization. Unsurprisingly based on the SDS-PAGE gel results with the proteins (Fig. 1d), only the seven mutant variant yielded protein crystals. We obtained a 1.76 Å crystal structure of the stabilized Cα/Cβ subunit (Supplementary Table S2). The side chains for all seven mutants are resolved in the structure and provide an indication of how the mutations are stabilizing the protein (Figs. 2 and 3).

In the crystal structure and the Rosetta model, phenylalanine 139 (**Cα S139F**) fills a partially buried cleft in the structure and forms tight packing interactions with the sidechain of Cα 129Q.

Notably, the orientation of the interaction leaves the polar groups on the glutamine open for forming hydrogen bonds with water. Despite being solvent exposed, **Cβ D155P** has the largest impact on protein stability of the seven mutations. The Rosetta energy calculations favor the proline because the residue has backbone torsion angles ($\phi = -73.4°$, $\psi = 52.3°$) compatible with the closed ring of the proline. One benefit of mutating to proline is that proline is more restrictive than aspartic acid in Ramachandran space and therefore loses less entropy when the protein folds[24]. Isoleucine 150 (**Cα T150I**) packs tightly with surrounding residues and removal of the threonine does not leave any buried polar groups without a hydrogen-bond partner.

**Cα A190T** is at the beginning of a turn in a solvent-exposed loop. In addition to increasing solubility on the surface, Rosetta predicted that this threonine would form a stabilizing hydrogen bond to Cα 129Q. However, the conformation of Cα 129Q appears to be influenced away from the hydrogen bonding conformation by another mutation, Cα S139F. Instead, the crystal structure shows the threonine adopting a different rotamer to form a hydrogen bond with the backbone on the other end of the loop's turn (backbone nitrogen of Cα 193N). Rosetta predicted that **Cβ S170D** would form a bidentate hydrogen bond across the interface with the sidechain of Cα 171R. The crystal structure shows Cβ S170D hydrogen bonding with the backbone nitrogen atom of Cα 171R (which was not previously participating in a hydrogen bond) and allowing the arginine's sidechain to become more solvent-exposed. **Cβ H139R** creates an interface-spanning hydrogen bond with Cα 124D. Rosetta modeled the arginine and aspartic acid to form a bidentate hydrogen bond, however the crystal structure shows that they form just one hydrogen bond so that the arginine can better pack with neighboring residues. The designed lysine at residue 134 on the β chain (**Cβ E134K**) has high b-factors and is likely interacting with several negatively charged amino acids in the vicinity.

After seeing the sidechain conformations in the crystal structure deviate from the Rosetta model for many of the designed polar interactions, we decided to investigate why Rosetta did not predict the observed sidechain conformations. One possibility is that the sidechain and backbone sampling protocol in Rosetta was unable to sample these conformations. Alternatively, Rosetta may have sampled them but predicted them to be higher in energy. To test between these hypotheses, we used the backbone coordinates from the crystal structure of the stabilized mutant as the starting point for fixed-backbone sidechain prediction simulations. The backbone conformation in our crystal structure is similar (0.338 Å rmsd) to the crystal structure of the wild-type protein, but there are small deviations that may affect sidechain positioning and their ability to form hydrogen bonds. Indeed, this is what we observed, given the crystal structure backbone Rosetta correctly predicted most of the sidechain conformations of the mutants (Supplementary Fig. S2). This result suggests that Rosetta did not correctly predict the sidechain conformations of these mutations in the original design model because it did not sample and/or favor the small backbone perturbations observed in the crystal structure.

Overall, the stability measurements and the crystal structure of the stabilized variant highlight some of the strengths and weaknesses of protein modeling. In general, it is more straightforward to predict the conformation and energetic consequences of hydrophobic mutations in the protein core. The constrained environment as well as the more limited flexibility of hydrophobic side chains allows more accurate prediction of the most favorable rotamer[25]. Polar amino acids are typically exposed to solvent and in general are longer with more sidechain degrees of freedom. Given the challenges associated with modeling polar residues, making use of information in a

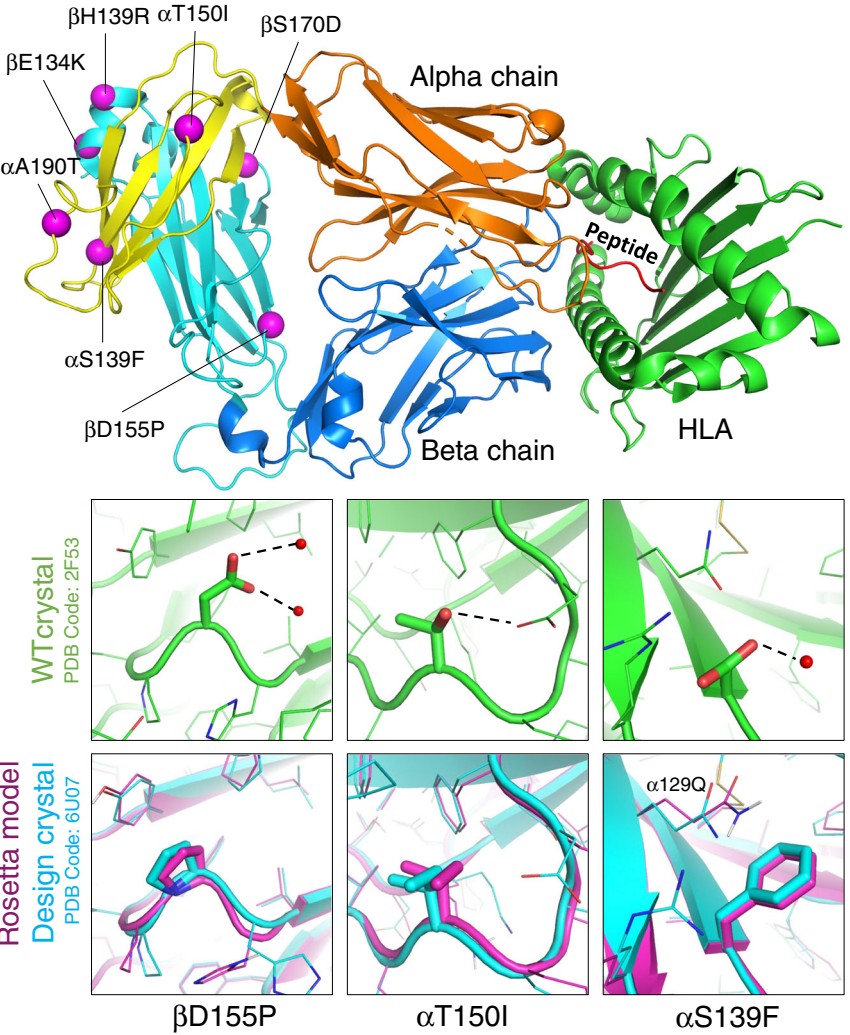

**Fig. 2 Overview of seven mutations and structural characterization of mutations to nonpolar amino acids.** (Top) here we show the location of each of the seven mutations with respect to the structure as a whole. The alpha chain is shown in orange and the beta chain is shown in blue, with the constant domains tinted a lighter hue on the left edge of the figure. (Bottom) three of the seven mutations were from polar side chains to hydrophobic side chains. Each column visualizes how the wild-type conformation has changed, showing the wild type in green (pdb: 2F53, energy minimized with Rosetta) and overlaying the Rosetta model (purple) with our crystal structure (blue).

MSA may be most useful when mutating polar amino acids. Consistent with this conclusion, all five of the stabilizing mutations that involved mutation to a polar amino acid were present in the MSA, while the stabilizing mutations that were not in the MSA involved mutations to hydrophobic amino acids. The absolute value of the Rosetta Energies (dE) for each of the stabilizing polar mutations was smaller than all the dEs of the mutations chosen without the requirement of being within the MSA. Without the MSA filter, most of the stabilizing polar mutations would not have been included in the group of 50 mutants that were screened.

**Impact of the Cα/Cβ designs on full-length TCRs.** We evaluated the impact of the Cα/Cβ designs on the expression and stability of unrelated, full-length, soluble TCRs. First, we generated the 1G4_122 anti-NY-ESO-1 TCR with and without the Cα/Cβ designs. Without the CαT166C/CβS173C disulfide, the 1G4_122 TCR expressed poorly in mammalian cells, but with the Cα/Cβ designs, the expression increased over 10-fold and the $T_m$ measured by DSC was 8 °C higher (Table 1, Fig. 4a). We also expressed and characterized three different TCRs targeting MAGE-A3 (pdb 5BRZ; NCBI: α=ABY74337.1/β=ACZ48691.1),

HIV (pdb 3VXT; NCBI: α=ABB89050.1/β=ACY74607.1), and WT-1 antigens[26–28]. The expression of the non-stabilized TCRs was poor (59 ± 32 mg/L), while the expression of the stabilized TCRs was significantly improved (280 ± 180 mg/L) (Table 1). For comparison, the average expression of seven different monoclonal antibodies (mAbs), when expressed simultaneously with the TCRs, was 235 ± 280 mg/L. The assembly of two of the TCRs (anti-WT-1 and anti-HIV) was poor based on the lack of distinct bands near 50 kDa expected for differentially glycosylated α/β TCRs, while all TCRs showed proper assembly in the presence of the stabilizing designs. (Supplementary Fig. S3a). DSF analysis of the TCRs demonstrated that stabilized TCRs all had a dominant unfolding $T_m$ between 60 and 70 °C, while the non-stabilized TCRs either demonstrated no clear unfolding event or an unfolding $T_m$ < 60 °C (Supplementary Fig. S3b). When comparing the expression of the TCRs stabilized only by the CαT166C/CβS173C or by the seven mutant designs, in all cases, the seven mutant designs described here significantly improved TCR expression (by several-fold) over the addition of the CαT166C/CβS173C disulfide (Table 1). While the stabilization of the NY-ESO-1 TCR by the seven mutant designs versus the CαT166C/CβS173C disulfide appeared to be comparable, the other three

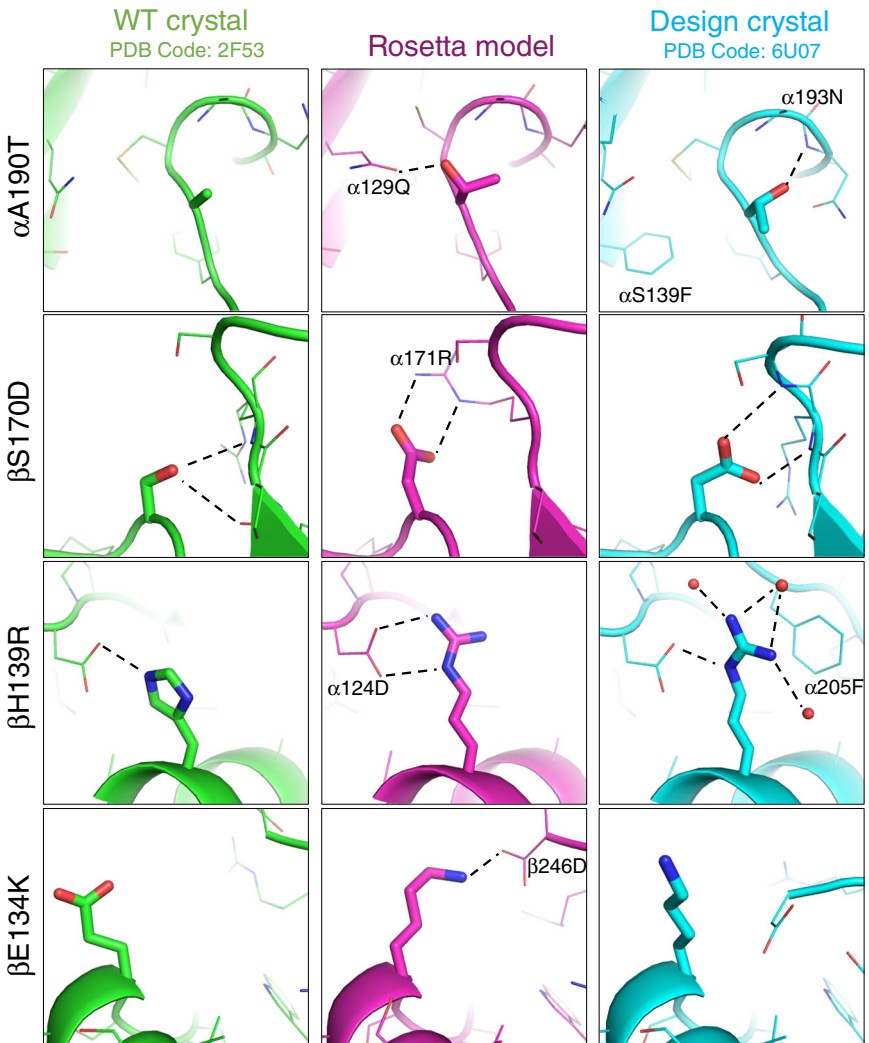

**Fig. 3 Structural characterization of mutations to polar amino acids.** The wild-type crystal structure (pdb: 2F53, energy minimized with Rosetta) is shown in green, The Rosetta models for the mutations is shown in magenta, and the crystal structure of the stabilized TCR Cα/Cβ (pdb: 6U07) with the seven stabilizing mutations is shown in cyan.

TCRs demonstrated superior stabilization using the seven mutant designs (Table 1).

The seven mutant stabilizing Cα/Cβ designs were also evaluated in combination with the CαT166C/CβS173C disulfide. Adding the seven mutant designs to the CαT166C/CβS173C disulfide resulted in a roughly 3-fold increase in titer and an increase in thermal stability by DSC for all four TCRs (Table 1, Fig. 4b). Interestingly, all four TCRs were more compact as assessed by SDS-PAGE analysis and analytical size exclusion chromatography (SEC, Fig. 4c, d).

To investigate further the more compact nature of the TCRs in the presence of the stabilizing Cα/Cβ mutations, we performed both hydrogen deuterium exchange (HDX) analyses and glycan occupancy studies using peptide mass mapping with the WT-1 TCR (CE10) with and without the stabilizing Cα/Cβ designs. HDX patterns of the CE10 TCR with and without the stabilizing Cα/Cβ mutations was nearly identical except for significant regions of the Cα chain (Fig. 4e). HDX protection of the Cα domain upon stabilization was not localized to the sites of mutation, but more globally to various regions of the Cα fold, including protection at the C-terminus. The wild-type and stabilized CE10 TCR was also evaluated for the presence/ occupancy of N-linked glycans before and after enzymatic N-

linked and O-linked deglycosylation. A single N-linked glycosylation motif exists in the Vα domain, three in the Cα domain, and one in the Cβ domain. Cα/Cβ stabilization led to a significant reduction in N-linked glycosylation within the CE10 TCR including within Vα, suggesting that stabilizing Cα/Cβ also stabilizes the Vα/Vβ domains (Table 2, Fig. 4f). O-linked glycosylation was probable at the C-terminus of Cα since the C-terminal peptide could not be resolved in the TCR lacking the stabilizing designs. A well-resolved C-terminal peptide lacking O-linked glycosylation was identified upon stabilization of the Cα/Cβ subunit. The Cα C-terminal region is variably present in different TCR crystal structures and absent in the NY-ESO-1 structure we used for the design. In the structure of the stabilized Cα/Cβ subunit, the Cα C-terminus is well resolved. Interestingly, the Cβ H139R mutation forms charge-charge interactions as well as a potential π-stacking interaction across the Cα/Cβ interface that may stabilize/lock-down the conformation of the Cα C-terminus, which also shows enhanced protection from HDX. Thus, the universal decrease in hydrodynamic radius observed for all the stabilized TCRs by SDS-PAGE and SEC was confirmed as a reduction in glycosylation.

Next, we assessed the ability of the stabilizing mutations to enable full-length TCRs to withstand variable domain mutation.

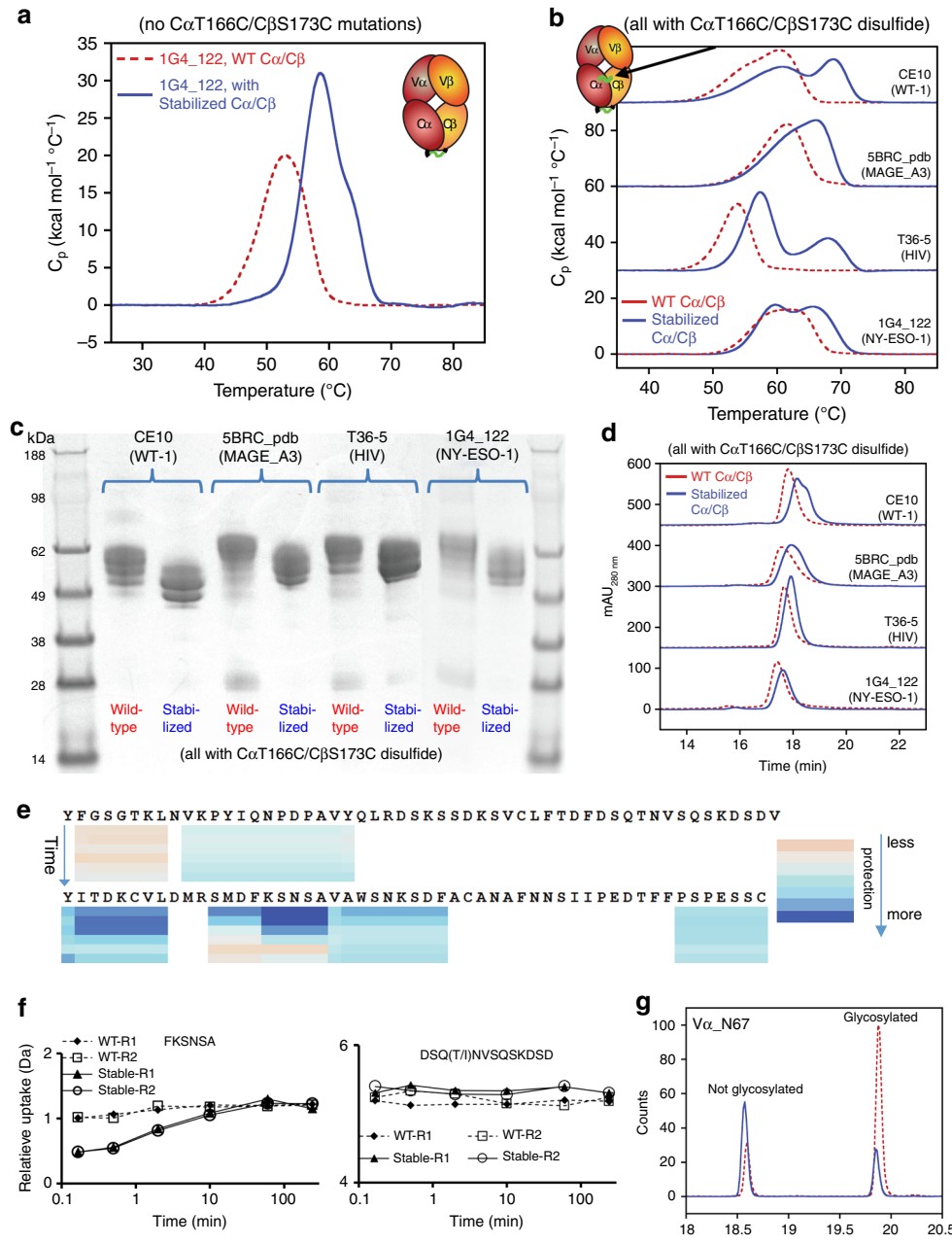

**Fig. 4 Characterization of four diverse α/β TCRs with and without a stabilized Cα/Cβ subunit. a** DSC curves of the 1G4_122 TCR with zero (WT, dotted red line) or seven (solid blue line) stabilizing Cα/Cβ mutations in the absence of the α166/β173 stabilizing disulfide. **b** DSC curves, (**c**) non-reduced SDS-PAGE, and (**d**) analytical SEC of four, diverse TCRs with zero (WT, dotted red line) or seven (solid blue line) stabilizing mutations all in the presence of the α166/β173 disulfide. **e** The only significant areas of backbone protection from deuterium exchange were in Cα. The heat map under each peptide region indicates the level of protection observed at 10 s, 30 s, 2 min, 10 min, 1 h, and 4 h. Lack of a heat map indicates no significant differences in deuterium exchange observed in the region. **f** Deuterium uptake plots of peptides from the wild-type CE10 TCR and stabilized CE10 TCR. The experiment was performed in duplicate and both results of the two experiments are shown on the plots denoted as replicate 1 (R1) or replicate 2 (R2). **g** Extracted ion chromatograms of the peptide containing the Vα_N67 N-linked glycosylation site. The peptide at 18.6 min was not glycoylsated while the peak 19.9 min shows up after enzymatic deglycosylation/conversion to Asp. The peptide from the wild-type TCR is a red dotted line and the peptide from the stabilized TCR is a solid blue line. Source data are available in the Source Data file.

Using Rosetta, a limited number of mutations in both the Vα and Vβ domains were generated in the 1G4_122 anti-NY-ESO-1 TCR. Ultimately, one (βN66E) was stabilizing, multiple mutations were neutral, and one was significantly destabilizing (βS49A, Fig. 5a). The Vα and Vβ mutations were combined and this small library of TCR variants were expressed in the absence and presence of the stabilizing Cα/Cβ mutations. A wide range of $T_m$s

were observed by DSF for the 1G4_122 TCRs in the presence of a wild-type Cα/Cβ subunit, while the stability of the 1G4_122 TCRs was virtually constant when the stabilizing Cα/Cβ subunit was present (Fig. 5b, c). A slightly higher apparent $T_m$ of the non-stabilized anti-NY-ESO-1 TCR containing βN66E versus that of the TCR containing the seven mutant designs is unlikely a real difference, but instead the result of the overlap of the unfolding

**Table 2 Glycan occupancy of the CE10 TCR in the absence and presence of the stabilizing Cα/Cβ designs. Occupancy was determined based on chromatography peak area of occupied/unoccupied peptides (e.g., Fig. 4g).**

| N-linked glycosylation site | %Glycan occupancy wild-type CE10 TCR | %Glycan occupancy Cα/Cβ stabilized CE10 TCR |
|---|---|---|
| Vα_N67 | 78 | 30 |
| Cα_N151 | 70 | 20 |
| Cα_N185 | 65 | 5 |
| Cα_N196 | n.d.[a] | 80 |
| Cβ_N186 | 60 | 25 |

[a]Peptide could not be resolved. Possible O-linked glycosylation at CαT204 occludes peptide resolution.

transitions of the variable and constant domains within the non-stabilized TCR making it difficult to accurately measure the variable domain $T_m$. As observed with the TCRs described above, the entire library of 1G4_122 variants achieved a roughly 3-fold increase in titer with the Cα/Cβ designs present (Fig. 5d). Interestingly, the $T_m$s of the 1G4_122 variants ranged from 49 to 65 °C, whereas the Cα/Cβ stabilized library all had $T_m$s tightly clustered around 62 °C (Fig. 5e) and were impervious to variable domain instability. These results suggest the stabilized Cα/Cβ subunit may rescue the expression and stability of TCRs with low intrinsic stability.

**Producing soluble TCR/CD3 bispecifics**. We believe a major benefit to stabilizing α/β TCRs would be the additional bispecific formats that could be enabled. First generation TCR bispecifics combine the specificity of soluble α/β TCRs with an anti-CD3 scFv in the ImmTac format[5]. This format enables exquisite redirected lysis potency; however, the major drawbacks from our perspective are what we perceive to be low yield bacterial expression and refolding as well as rapid serum clearance that makes in vivo dosing challenging. With the significant increase in expression and proper folding of α/β TCRs in mammalian cells afforded by the stabilizing the Cα/Cβ subunit, we believe additional bispecific formats should be accessible that include moieties with enhanced pharmacokinetics (PK) as well as avidity to the tumor cells.

We generated both IgG-like and Tandem bispecific antibodies (BsAbs, Fig. 6a). The BsAbs utilized the 1G4_122 α/β TCR that binds the HLA-A2/NY-ESO-1 peptide complex and was engineered to bind with high affinity (1 nM)[21]. The IgG format requires the expression of two heavy chains that require an antibody Fc heterodimerization design to achieve proper HC assembly. The previously published 7.8.60 heterodimer design was chosen[29]. We expressed the IgG BsAb with a wild-type or stabilized Cα/Cβ subunit. As we observed with the soluble TCRs, the IgG BsAb containing a stabilized Cα/Cβ subunit expressed over two-fold higher than the construct with a wild-type Cα/Cβ subunit. The IgG BsAb protein lacking the stabilizing mutations was found to be a mixture of TCR/CD3 BsAb and anti-CD3 half-antibody (Fig. 6b). The anti-CD3 half/antibody was observable as a low-molecular weight (LMW) peak by analytical SEC after IgG-Fc affinity purification (Fig. 6b). Both the IgG and Tandem BsAbs with a stabilized Cα/Cβ subunit were secreted as >90% monodisperse (i.e., pure) proteins coming directly from the cell culture based on their profile after a single affinity chromatography step (Fig. 6b). The correct ratio of each of the chains and correct molecular weight of the fully assembled

BsAbs was confirmed using SDS-PAGE (Fig. 6c). The purity of the stabilized TCR/CD3 BsAbs directly off an affinity chromatography step is remarkable, even for antibody-based bispecifics[30]. Both the improved expression and optimal purity would have a big impact on translatability to the clinic and for commercializability of the platform since the cost of biologics manufacturing is high and the removal of product-related impurities can be challenging and result in significantly reduced overall product yields.

The BsAbs were next evaluated for their function. Flow cytometry was performed with Jurkat cells to assess CD3 binding and with 624.38 and Saos-2 tumor cells to assess peptide/MHC binding. Both the 624.38 melanoma line and the Saos-2 sarcoma line have been characterized to express HLA-A2[31]. The TCR/CD3 bispecifics demonstrated insignificant binding in the absence of endogenously added NY-ESO-1 peptide (SLLMWITQC), but strong binding to these cells once the NY-ESO-1 peptide was added to the tumor cells (Fig. 7a). Antigen peptide pulsing is often required induce recombinant T cell-mediated killing of tumor cells in vitro[31,32]. After adding synthetic NY-ESO-1 peptide to the tumor cells, both the IgG and Tandem BsAbs bound both the 624.38 and Saos-2 lines (Fig. 7a). The Tandem BsAb appeared to bind slightly better to the two separate HLA-A2+ tumor cells compared to the IgG BsAb. Both BsAbs showed a 10- to 30-fold decrease in binding potency to Jurkat cells compared to the bivalent anti-CD3 mAb due to a loss of avidity. This is a commonly observed phenomenon when converting 2-arm binding mAbs to a 1-arm binding mAb or Fab. Given the weaker binding to tumor cells observed for the IgG BsAb, a couple higher potency TCR[21] versions of the IgG BsAb were generated to offset the potency loss (Supplementary Fig. S4a). The 1G4_107 TCR variant demonstrated improved potency binding to tumor cells over the 1G4_122 variant by flow cytometry (Supplementary Fig. S4b).

Next the BsAbs were assessed for their ability to redirect T cells to kill the HLA-A2+ tumor cell lines. The BsAbs were titrated onto the 624.38 and Saos-2 tumor cells in the absence and presence of the synthetic NY-ESO-1 peptide. Next, unstimulated primary T cells were added to the cells. Neither BsAb induced T cell redirected lysis in the absence of the NY-ESO-1 peptide. This was not entirely unexpected as primary T cells expressing recombinant TCR targeting NY-ESO-1 have been shown to require peptide pulsing to induce redirected lysis[31]. Upon addition of the peptide, both BsAbs demonstrated the ability to redirect primary T cells to kill the tumor cells. The TCR/CD3 IgG BsAb demonstrated modest potency (~1 nM EC50) and approximately 50% killing on Saos-2 cells and showed no activity on 624.38 cells over the 48 h of the assay (Fig. 7b). Increasing the affinity of the TCR/CD3 IgG BsAb using a known high affinity variant[21] did not increase its potency or saturated level of killing (Supplementary Fig. S4c). The tandem BsAb format, however, was significantly more potent (~0.03 and 0.1 nM EC50s) and effective (approximately 90% and 40% killing) on Saos-2 and 624.38 cells, respectively, over the 48 h of the assay. (Fig. 7b).

To determine whether the tandem BsAb can redirect T cells to kill HLA-A2+ tumor cells without pulsing the NY-ESO-1 peptide exogenously, we evaluated its ability to kill A375 HLA-A2+/NY-ESO-1+ melanoma cells since ImmTacs have been shown to kill this cell line in the absence of pulsed peptide[33]. The IgG-like BsAb was not evaluated since it demonstrated inferior T cell redirected lysis when pulsing peptide onto HLA-A2+ tumor cells. The affinity ($K_D$) of the 1G4_122 α/β TCR variant in the tandem BsAb was 1 nM[21], which is much lower than the affinity of the ImmTAC-NYE shown to kill tumor cells by McCormack and coworkers, which has a reported affinity of 50 pM[33].

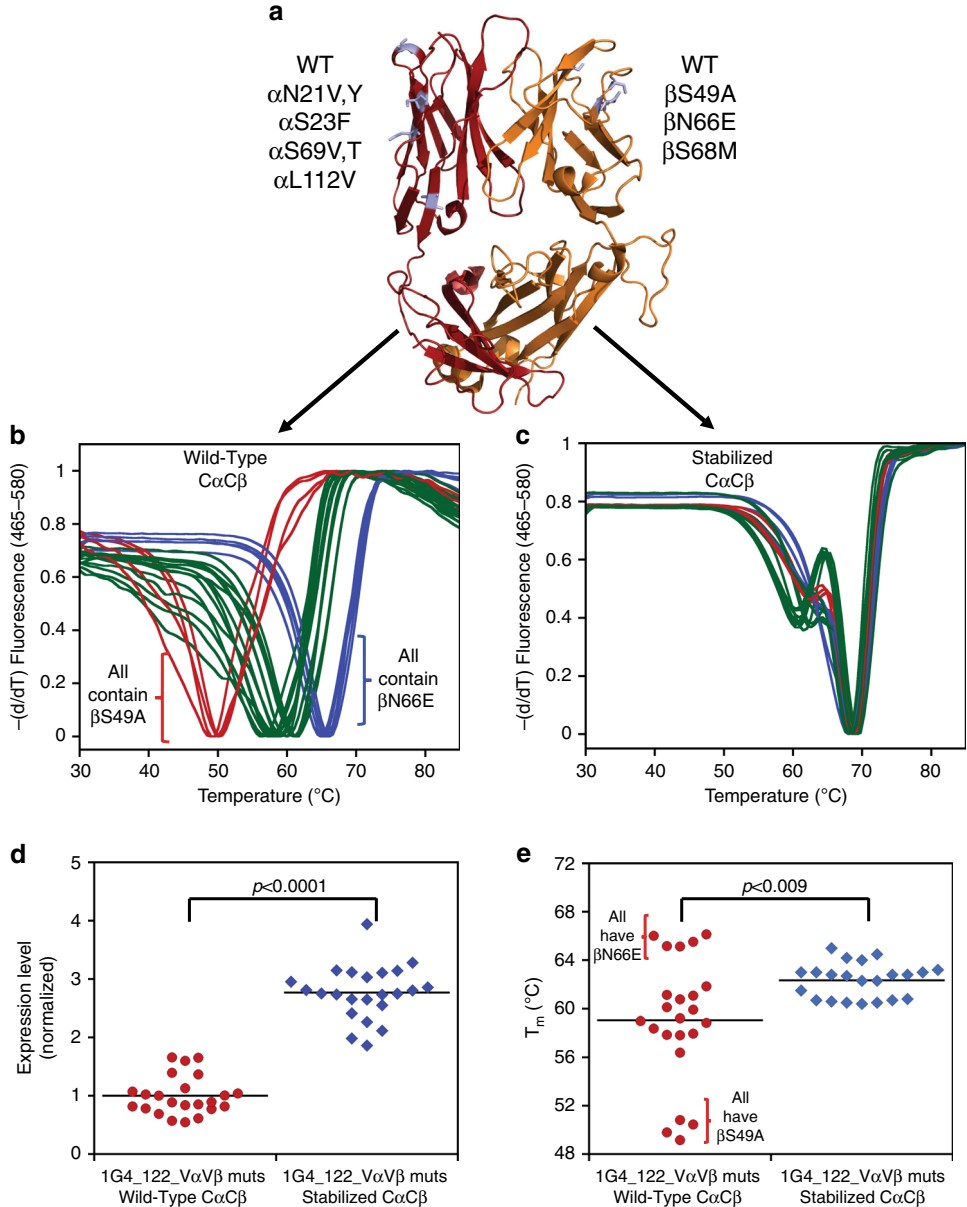

**Fig. 5 Characterization of stabilizing and destabilizing Vα/Vβ mutants within the 1G4_122 TCR with or without the stabilized Cα/Cβ subunit.**
**a** Cartoon depiction of the 1G4_122 (pdb 2F53) with various Vα/Vβ mutations shown in lavender. DSF curves of the 1G4_122 TCR in the presence of various Vα/Vβ mutant combinations and in the absence (**b**) and presence (**c**) of a stabilized Cα/Cβ subunit. Normalized expression levels (**d**) and lowest Tm (**e**) of the 1G4_122 TCR with different variable domain mutations in the absence (●) and presence (◆) of a stabilized Cα/Cβ subunit. Significant differences in expression and $T_m$ between TCRs with and without the seven stabilizing mutations were evaluated by calculating an exact P-value using a Wilcoxon Rank Sum/Mann-Whitney two-tailed test assuming unequal variance and non-normal distributions between the groups compared. The horizontal line within plots (**d**) and (**e**) is the mean. Source data are available in the Source Data file.

Therefore, we produced a second tandem BsAb using the 1G4_113 α/β TCR variant with 26 pM affinity to the NY-ESO-1/HLA-A2 complex[21]. Both the 1 nM and 26 pM 1G4/anti-CD3 tandem BsAbs were titrated onto A375 melanoma cells in the presence of expanded primary T cells from two separate donors. Both tandem Fabs could potently kill the A375 cells expressing endogenous NY-ESO-1 in the absence of pulsed peptide (Fig. 7c). As observed with the IgG-like BsAbs in Supplementary Fig. S4b, both tandem Fabs showed similar redirected T cells lysis potency on A375 cells regardless of their affinity differences (Fig. 7c). Thus, the tandem BsAb bifunctionals can potently redirect T cells to kill tumor cell lines endogenously displaying the NY-ESO-1 peptide on HLA-A2.

## Discussion

Based on the data presented here, the stabilized Cα/Cβ subunit will likely improve the expression and stability of most recombinant TCRs including those with low stability variable domains. Compared to traditional bacterial methods of production in inclusion bodies followed by solubilization, refolding, and purification[7], the mammalian expression/secretion of fully assembled/oxidized TCRs described here should dramatically increase their ease of production. The stabilized Cα/Cβ subunit led to antibody-like expression levels for TCRs regardless of the presence of the known stabilizing disulfide bond[14]—an achievement not provided by the disulfide bond in isolation. This increased expression enables the robust production of various TCR/

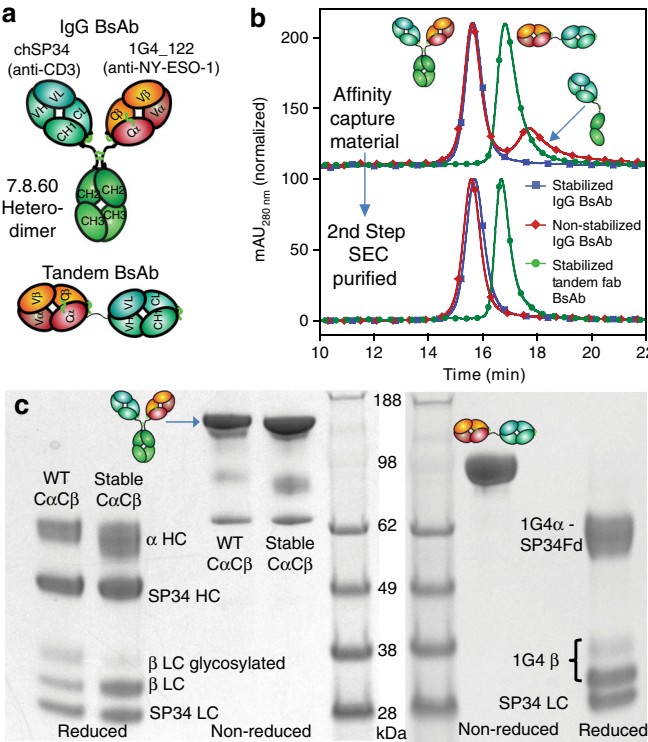

**Fig. 6 Characterization of IgG-like TCR/CD3 proteins. a** Schematic diagrams of the TCR/CD3 bispecific (BsAb) molecules. **b** Analytical SEC chromatogram and **c** Non-reduced and reduced SDS-PAGE of the IgG BsAbs using the wild-type NY-ESO-1 TCR and a stabilized NY-ESO-1 TCR as well as a tandem Fab BsAb using only the stabilized NY-ESO-1 TCR. Source data are available in the Source Data file.

antibody bifunctional molecules. An interesting possible application not evaluated in this report would be to include the stabilized Cα/Cβ subunit within full-length TCRs transgenically induced within primary T cells. Difficulty with chain association and expression have seen minor improvements via the introduction of the disulfide bond, the order of expression of the α/β chains, or introduction of hydrophobics in the transmembrane region[34,35]. It would be interesting to determine if the stabilized Cα/Cβ subunit described here would generally improve cell-surface TCR expression for cell based-therapy approaches.

The stabilized TCRs also displayed reduced overall glycosylation (both variable and constant domains) compared to their non-stabilized counterparts. Glycosylation has been known to stabilize and solubilize many different proteins including antibody Fc and, in specific cases, antibody Fab moieties[36]. Glycoengineering has even been used as an approach to stabilize proteins, reduce their propensity to aggregate/misfold, or improve their solubility[36]. The loss of glycosylation of Cα was shown to dramatically reduce the expression of IgG-like proteins containing a Cα/Cβ subunit[37]. Thus, we were surprised to find that the stabilizing designs significantly reduced the N-linked glycosylation by roughly 50% per site throughout the entire TCR. Putative O-linked glycosylation at the C-terminus of Cα was also absent in the stabilized form. We hypothesize that stabilizing interactions reduce the conformational fluctuation of these sites within the folding/folded protein, reducing access of the enzymes to decorate the canonical N-linked motifs and O-linked site(s). This includes an N-linked site within Vα, providing further evidence that stabilizing the Cα/Cβ subunit generally stabilizes the entire TCR. Further, full-elimination of the N-linked glycans on Cα via mutation had no impact on protein expression within the stabilized constructs.

Clear geometrical constraints exist for TCR/CD3 bifunctionals' ability to redirect T cells to lyse tumor cells. We show a substantial difference in overall activity and potency between the IgG and tandem BsAbs described here even though they contained identical soluble TCR and anti-CD3 arms. Such geometrical/size impacts on redirected lysis potency/activity have been demonstrated previously for antibody-based Tumor-Associated Antigen/CD3 BsAbs[38–40]. We are not aware of similar data being shown for TCR-based therapeutics. The flexibility and spacing of the soluble TCR and anti-CD3 Fab subunits appears critical to achieving high potency redirected lysis. Many discussions regarding the size of the antigen and or bispecific construct and their ability to fit within the immune synapse have been raised as a determining factor for their activity[41]. However, we observed differences between the IgG-like and tandem BsAbs in their ability to bind tumor cells in the absence of the T cells, which complicates the size/immune synapse interpretation. Indeed, enhancing the affinity of both the IgG and tandem Fab BsAb by 20- to 30-fold did not result in significantly improved potency, perhaps because the geometrical constraints of each format dictate the maximum potency that can be observed once a certain affinity is reached. The individual binding characteristics to each cell type may also play a big role in the case of these TCR/CD3 BsAbs. Overall, we believe the significant advantages afforded by the stabilized Cα/Cβ designs could be valuable not only for soluble TCR-based therapeutics, but could be leveraged to improve recombinant α/β TCR expression in cellular systems including those that use designed TCRs as part of cell based therapies.

## Methods

**Molecular biology.** NY-ESO-1 1G4_122 soluble TCR, and the individual NY-ESO-1 Vα/Vβ and Cα/Cβ subunits were originally generated using gBlocks (Integrated DNA Technologies, IDT) and subcloned into mammalian expression vectors (Lonza) using recombinase cloning (In-Fusion, Clontech). Sequences for 1G4_122,107, and 113 were derived using the 2F53 crystal structure[42] and the CDRs described elsewhere[21]. A murine kappa light chain leader sequence was used to drive secretion of each protein and 8xHistags were added recombinantly to the N-termini of the α-chains. DNA sequences were derived using IDT optimized codon algorithms.

Single mutant libraries were created using a site-directed mutagenesis protocol. Briefly, the protocol employs the supercoiled double-stranded DNA vector and two synthetic oligonucleotide primers (generated at IDT) containing the desired mutation(s). The oligonucleotide primers, each complementary to opposite strands of the vector, are extended during thermal cycling by the DNA polymerase (HotStar HiFidelity Kit, Qiagen) to generate an entirely new mutated plasmid. Following temperature cycling, the product is treated with Dpn I enzyme to eliminate the methylated, non-mutated plasmid that was prepared using E. coli (New England BioLabs). Each newly generated mutant plasmids is then transformed into Top 10 E. coli competent cells (Life Technologies). Colonies were picked, cultured, miniprepped (Qiagen), and sequence verified. Mutant combinations were generally synthesized as gBlocks.

TCR/CD3 constructs were created using overlapping PCR of each fragment and recombinase cloning. The chimeric SP-34 anti-CD3 heavy and light chain sequences were published previously[38].

**Protein expression and purification.** α/β constructs were co-transfected for transient expression in either 24 well plates (1 mL scale) or individual flasks (100–200 mL scale) as described previously[43]. Briefly, plasmids were transfected in CHO K1SV KO cells maintained in a DMEM-based medium with 8–12 mM L-Glutamine (LM-Growth, SAFC Cat#59202C-100). Cells were maintained in shake flasks at 37 °C, 6–8% $CO_2$ in an 80% humidified incubator. Transient transfection of plasmids was performed using Polyethylenimine Max (PEI Max from Polysciences Cat# 24765-2). CHO cells were seeded at 1.5e6 cells/mL. DNA (1.6 mg/L coding plasmid + 1.6 mg/L herring sperm DNA) and PEI Max (8 mg/L) were sequentially added to the cells. Cultures were maintained at 32 °C in an incubator with 6–8% $CO_2$ and 80–90% humidity with shaking at 250 rpm on a ¾ inch shaking diameter shaker. Secreted protein was harvested by centrifugation of the cell cultures for 5 min. Supernatants were passed through 2 μM filters for purification[44,45]. Small scale expressions of TCR and TCR fragments were quantified using Ni$^{2+}$-NTA tips (Pall Forte Bio) reading on the Octet RED384 System. Purified TCR Cα/Cβ and full-length α/β TCR proteins were used to generate standard curves.

TCR/TCR fragment His-tag proteins were purified using a Ni$^{2+}$ immobilized resin (cOmplete™ His-Tag, Millipore Sigma) and an AKTA Pure system (GE Healthcare). The column was equilibrated with at least 5 column volumes of

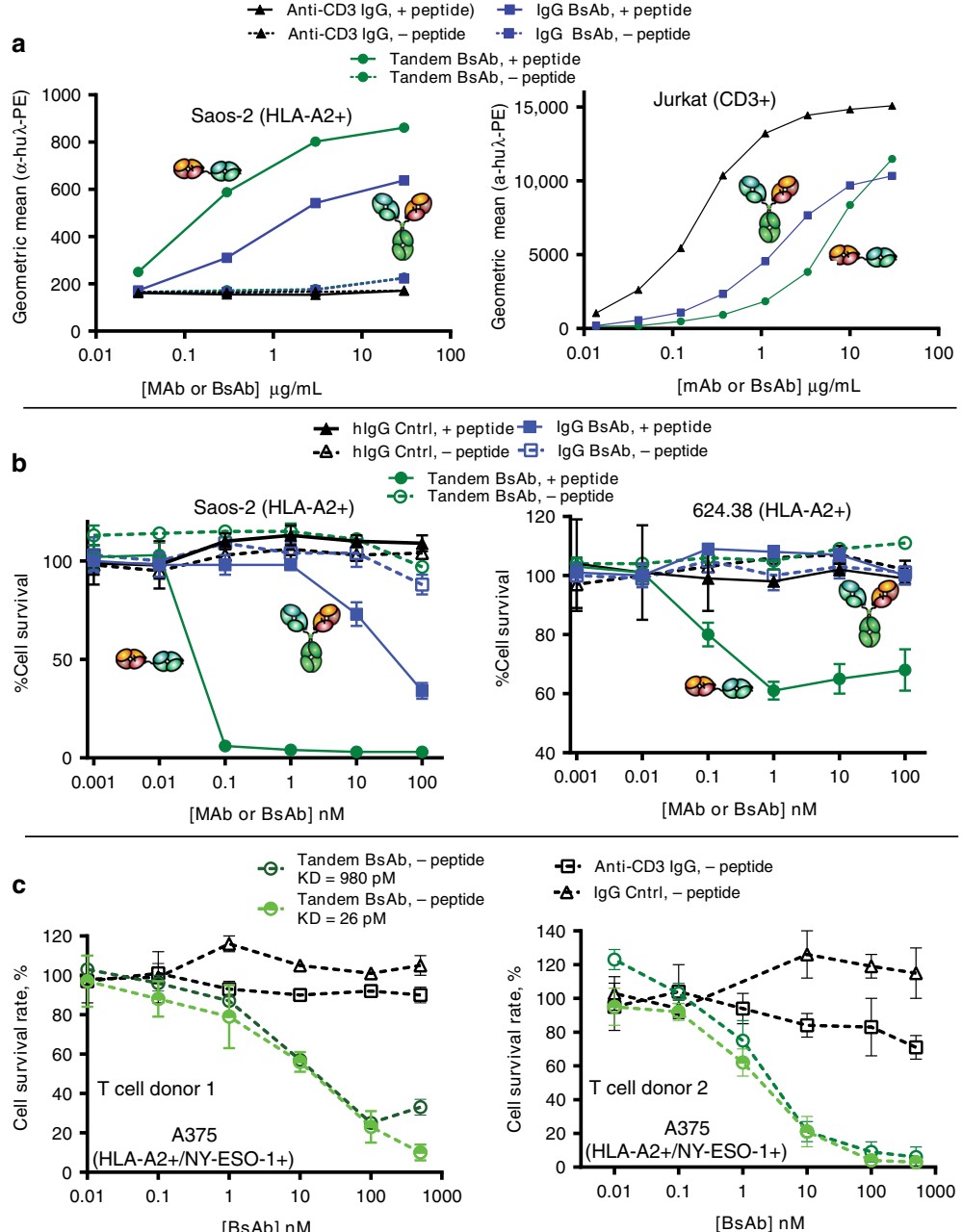

**Fig. 7 Cell binding and T cell redirected killing properties of the TCR/CD3 BsAbs. a** Flow cytometry cell binding titrations of the BsAb molecules with HLA-A2+ Saos-2 tumor cells with or without pulsing of the NY-ESO-1 peptide SLLMWITQC (left) and with CD3+ Jurkat cells (right). Each data point is the Geometric mean of the fluorescence of the cells labeled with the CD3 mAb or TCR/CD3 BsAb and secondary antibody. **b** T cell redirected killing of HLA-A2+ Saos-2 (left) and 624.38 (right) tumor cells facilitated by TCR/CD3 BsAbs with and without pulsing with exogenous NY-ESO-1 peptide. **c** TCR/CD3 tandem BsAb-mediated T cell redirected killing of A375 HLA-A2+/NY-ESO-1+ melanoma cells (no addition of exogenous peptide). Panels to the right and left use T cells expanded from different donors. For plots in sections (**b**) and (**c**), the data points are the mean of three replicates and the error bars are the standard deviation. Source data are available in the Source Data file.

binding buffer (20 mM Tris-HCl pH 8.0, 0.5 M NaCl) at a flow rate of 1 mL/min for 1 mL columns and 5 mL/min for 5 mL columns, respectively. After passing the CHO expression supernatant over the column to capture the His-tagged TCRs or TCR fragments, the TCR proteins were eluted with ten column volumes of elution buffer (20 mM Tris-HCl pH8.0, 0.5 M NaCl, 250 mM Imidazole). Imidazole was later removed by passing the proteins through a preparative size exclusion column (SRT-10C SEC300) or via dialysis against a phosphate buffered saline solution (PBS) at pH 7.2–7.4 using VIVASPIN 6 concentrators with a 10,000 MW cutoff. IgG-Fc containing proteins were purified using mAbSure resin (GE Healthcare) and preparative size exclusion. Sodium dodecyl sulfate polyacrylamide gel electrophoresis (SDS-PAGE) was performed using 4–12% Bis Tris gels according to the manufacturer (Life Technologies). Reductions and alkylations were performed with 1 mM dithiothreitol (DTT) and 1 mM N-ethylmaleimide (NEM).

**Differential scanning fluorimetry and calorimetry.** Protein unfolding was monitored on the Lightcycler 480II RT PCR machine (Roche) using SYPRO Orange dye (Invitrogen, S6651). Excitation and emission filters were set at 465 nm and 580 nm, respectively. The temperature was ramped from 25 °C to 95 °C at a rate of 1 °C/s. Stock solutions of protein were stored in running buffer (1 X PBS pH 7.4).

Final assay conditions were 0.2 mg/mL protein and a 500-fold Sypro dilution in running buffer. The experiment was performed by diluting the proteins to 0.4 mg/ml and diluting Sypro 250-fold, then combining the diluted protein and Sypro in equal parts in a 96-well plate (final volume: 30 μL per well). The samples were divided into 6 μL triplicates on a 384-multiwell assay plate (Roche, 04-729-749-001).

The midpoint of thermal unfolding ($T_m$) determination for each protein was performed on the LightCycler Thermal Shift Analysis Software (Roche) using the first derivative method. The analysis software smoothed the raw fluorescence data

and the $T_m$ was collected by determining the temperature where the upward slope of fluorescence vs. temperature was the steepest (i.e., temperature where first derivative of melt curve was maximal).

For DSC, thermal unfolding profiles were measured using an automated capillary DSC (capDSC, MicroCal/Malvern). Scans were performed at 1.5 °C/min using the low feedback mode. Protein concentrations were 1 mg/mL. The scan range was 20–110 °C. All 96-well plates containing protein were stored within the instrument at 6 °C. Unfolding transition midpoints ($T_m$s) were determined by fitting the curves using the non-two-state model with the Origin 7 software provided by the manufacturer.

**Rosetta energy predictions**. We used the TCR crystal structure with PDB code 2F53 and removed chains A–C, leaving only the alpha and beta chains. The remaining structure was relaxed using the FastRelax protocol in Rosetta with coordinate constraints (energy penalties that grow as a backbone atom deviates from its starting position) on every residue's CA atom[46]. All possible mutations (except for mutations to cysteine) were also simulated using the FastRelax protocol. FastRelax consists of two sub-protocols: (1) fixed-backbone rotamer substitutions (i.e., sidechain conformation sampling) and (2) all-atom minimization of backbone and sidechain torsion angles. The first sub-protocol generates a rotamer library for each residue position and stochastically samples rotamers at user-allowed positions. The lowest-energy combination of rotamers is assigned to the protein after this sampling is repeated many times (hundreds of thousands). The second sub-protocol uses gradient-based minimization of torsion angles to relax the structure into its local minimum in the Rosetta energy landscape. We used the energy function REF2015 for this study[47,48]. Certain considerations were made to prevent noise in the Rosetta energy predictions. For FastRelax's first sub-protocol, we only allowed residues within 10 Å of the mutation to sample alternate sidechain conformations. We added coordinate constraints to all residue positions in the second sub-protocol to prevent the backbone from varying much from the starting structure. After FastRelax completed, we allowed one final run of all-atom minimization without coordinate constraints; allowing the protein to relax into its local, unconstrained energy minimum. Our Rosetta script is provided in the supplementary information. We ran ten independent trajectories for every possible mutation and used the average score of the three lowest-scoring trajectories as the representative score for that mutation. The version identifier (git sha1) of our Rosetta version was 360d0b3a2bc3d9e08489bb9c292d85681bbc0cbd, released on 4 June 2017.

**Multiple sequence alignment**. We assembled a list of 39 CA and 53 CB sequences by hand-curating a MSA of TCR sequences from various organisms. We assembled two lists of TCR-similar sequences by passing the sequences of the CA and CB chains of the 2F53 structure into BLAST using NCBI's database of non-redundant protein sequences (169,859,411 sequences in database), allowing only sequences from animals. Both lists were purged by hand of any sequences that did not appear to be a TCR either because of poor alignment or because of the sequence's label. We partitioned the list of mutations deemed stabilizing by Rosetta into two lists: (1) mutations present in the MSA and (2) mutations absent from the MSA. We selected the 30 mutations from list 1 and 25 mutations from list 2 for experimental testing. Selection decisions were made primarily by Rosetta score but some candidates were removed for the sake of increasing design diversity. To aid the decision process we made use of a figure that displayed two computational metrics for all possible single point mutants in the constant domains (Supplementary Fig. S1): the calculated Rosetta score for the mutant and a log odds score based on the mutation representation in the MSA. By splitting the mutations into two lists, we could promote mutations that were deemed lower-tier by Rosetta if they have been shown to be compatible with any T-Cell Receptor structure.

**Modeling Cα/Cβ mutant combinations**. After the best performing point mutations were determined experimentally, we ran Rosetta simulations to verify that each mutation was compatible with every other mutation. We separately applied each mutation to the pre-relaxed 2F53 structure and ran fixed-backbone sidechain repacking. We performed the same simulations with every pair of mutations and ensured that the Rosetta energy of the pair was not dissimilar to the sum of the energies of the mutations individually.

**Protein crystallization**. Purified protein was crystallized at 8 °C using the sitting-drop vapor-diffusion method. Crystals were obtained by mixing one part protein solution at 15 mg/mL (10 mM Tris-HCl pH 7.5, and 150 mM sodium chloride) with one part reservoir solution containing 23% PEG 4 K, 300 mM magnesium sulfate, and 10% glycerol. Drops were immediately seeded with crushed crystals grown from 15% PEG 3350, and 100 mM magnesium formate. Single, plate-shaped crystals were observed within one week. These were harvested into a drop of reservoir solution containing 15% glycerol, and flash frozen in liquid nitrogen.

**X-ray data collection and structure determination**. Synchrotron X-ray diffraction data were collected on a single crystal at the Advanced Photon Source on beamline 31-ID-D (LRL-CAT). The resulting data were integrated using autoP-ROC[49], and merged and scaled with SCALA[50]. The structure was solved by Molecular Replacement with Phaser[51], using the TCR constant domains of a previously solved TCR-pMHC complex (Protein Databank accession code 2F53) as the search model[42]. Several rounds of model building and refinement were conducted with COOT[52] and REFMAC5[53], as implemented in the CCP4 software package v. 7.0[54]. The final model was validated with MolProbity[55]. Additional data collection and refinement details can be found in Supplementary Table S2 of the Supplementary Information.

**Peptide mapping**. The wild-type and stabilized CE10 TCRs were denatured, reduced, cysteine-alkylated, and digested for the analyses. Briefly, 100 μg of TCR variant was denatured in 6 M guanidine prior to reduction with 10 mM dithiothreitol at 37 °C for 30 min. Next, 15 mM iodoacetamide was added and the reaction was allowed to proceed for 45 min in the dark. Each sample was then buffer-exchanged into 10 mM TRIS, pH 7.5 using spin filters. 5 μg of trypsin was added to the reduced/alkylated sample and incubated for 4 h at 37 °C. The trypsinized samples were divided into two equal parts. The first sample was deglycosylated by adding 1 μL PNGaseF (ProZyme) and 5 μL 10x digestion buffer and incubating overnight at room temperature, while the second sample was left untreated.

To determine the glycan occupancy, an aliquot of PNGase F treated and untreated tryptic digest was injected onto an Agilent 1290 UPLC coupled to an Agilent 5210 QTOF Mass Spectrometer for peptide mapping and mass alignment. Briefly, 5 μg TCR digest was injected on to a Waters 150 mm × 2.1 mm 1.7 μm C18 CSH UPLC column and gradient separated from 2% B (0.1% formic acid in acetonitrile) to 35% B over 30 min. The eluting peptides were injected into the mass spectrometer for mass analysis with a scan range from 200 to 2000 $m/z$, source temperature of 350 °C, capillary voltage of 4 kV, fragmentor at 250 V, capillary exit at 75 V, source gas at 40 PSI, and cone gas at 12 PSI. The peptide mass spectra was aligned to the protein sequence by Mass Hunter/Bioconfirm 7.0 software with a mass accuracy of 5 ppm resulting in over 90% sequence coverage. To measure glycan occupancy, deamidation of Asn was added as a variable modification and each predicted glycan was checked manually by extracted ion chromatogram (EIC) with 5 ppm mass accuracy. The ratio of Asn to Asp was calculated using the EIC and reported as percent glycan occupancy: Asn being unoccupied and Asp being occupied since the enzymatic deglycosylation converts Asn to Asp.

**Hydrogen/deuterium exchange coupled with mass spectrometry**. HDX-MS experiments were performed on a Waters nanoACQUITY system with HDX technology[56], including a LEAP HDX robotic liquid handling system. The deuterium exchange experiment was initiated by adding 55 μL of $D_2O$ buffer containing 0.1x PBS to 5 μl of each protein (concentration was 25 μM) at 15 °C for various amounts of time (0 s, 10 s, 1 min, 10 min, 60 min, and 240 min). The reaction was quenched using equal volume of was 0.32 M TCEP, 0.1 M phosphate pH 2.5 for 2 min at 1 °C. 50 μL of the quenched reaction was injected on to an on-line pepsin column (Waters BEH Enzymate) at 14 °C, using 0.2% formic acid in water as the mobile phase at a flow rate of 100 μL/ min for 4 min. The resulting peptic peptides were then separated on a C18 column (Waters, Acquity UPLC BEH C18, 1.7 μm, 1.0 mm × 50 mm) fit with a Vanguard trap column using a 3–85% acetonitrile (containing 0.2% formic acid) gradient over 10 min at a flow rate of 50 μL/min. The separated peptides were directed into a Waters Xevo G2 time-of-flight (qTOF) mass spectrometer. The mass spectrometer was set to collect data in the $MS^E$, $ESI^+$ mode; in a mass acquisition range of $m/z$ 255.00–1950.00; with a scan time of 0.5 s. The Xevo G2 was calibrated with Glu-fibrinopeptide prior to use. All acquired data was mass corrected using a 2 μg/ml solution of LeuEnk in 50% ACN, 50% $H_2O$ and 0.1% FA at a flow rate of 5 μl/min every 30 s ($m/z$ of 556.2771). The peptides were identified by Waters Protein Lynx Global Server 3.02. The processing parameters were set to low energy threshold at 100.0 counts, an elevated energy threshold at 50.0 counts and an intensity threshold at 1500.0 counts. The resulting peptide list was imported to Waters DynamX 3.0 software, with threshold of 5 ppm mass, 20% fragments ions per peptide based on peptide length. The relative deuterium incorporation for each peptide was determined by processing the MS data for deuterated samples along with the non-deuterated control in DynamX 3.0 (Waters Corporation).

**T cell redirected lysis assays**. Saos-2 and A375 cells were from ATCC (Cat#HTB-85 and CRL-1619, respectively) and the 624.38 cells were from the NCI/NIH DTP, DCTD Tumor Repository[57]. All three lines were cultured in Complete Media (RPMI 1640 /10% fetal bovine serum (FBS) Corning/gentamicin—Gibco). Primary naïve T cells were from StemExpress (Cat#PB03020C, Lot#1804170175; Donor#D001003581). For the assay, the tumor cells were removed from their culture flask using Accutase (Innovative Technologies) and spun down for 10 min at 170g. The tumor cells were resuspended in Complete Media and seeded at 5000 cells/well in 96-well black, clear bottom plates (Perkin Elmer) and SLLMWITQC (NY-ESO-1) peptide (synthesized by CPC Scientific), 100 μL total volume, for 3 h. After 3 h, 50 μL of the peptide solution was removed from each well. TCR/CD3 bifunctionals (50 μL/well at 4×) or control antibodies (in-house, recombinantly-produced anti-CD3 chimeric SP34[38] and irrelevant anti-CD3 IgG1[58]) were then titrated onto the cells using 1:10 serial dilutions starting at 200 nM (2×) and ending at 0.002 nM (2×) in triplicate for 30 min. During this period, primary T cells were thawed and washed twice in Complete Media and gentamicin. T cells (100 μL) were then added at 50 K cells/well (200 μL total volume). After the T cells were added the

TCR/CD3 bifunctionals were 1×(starting at 100 nM and ending at 0.001 nM) and the peptide was at 1.5 µg/ml for the duration of the experiment. Non-peptides plates were treated the same as those with peptide except for the absence of peptide in the procedure. Cells were incubated for 48 h at 5% $CO_2$ and 37 °C. At 48 h, the plates were washed gently (twice) with serum free RPMI 1640. For assays with the A375 cells, primary T cells (Donor 1: StemCell Cat#70024, Lot#190981591C, Donor 110044380; and Donor 2: StemExpress Cat#PB03020C, Donor 1804170165) were thawed and resuspended in Complete Media (RPMI 1640 /10% FBS Corning/gentamicin—Gibco) containing anti-CD28 (BD Biosciences 555725, CD28.27, 2.5 µg/mL) and IL-2 (R&D systems 202-IL/CF, 2 ng/mL) and expanded by being placed in flasks precoated overnight with anti-CD3 (BD Biosciences Cat#555329, UCHT1, 5 µg/mL). T cell expansion was allowed to proceed between 4 and 14 days before use. At day 7, T cells were re-stimulated by transfer into a new flask pre-coated with UCHT1 and containing fresh Complete Media supplemented with anti-CD28 and IL-2. 100 K expanded T cells/well were washed 2 times in Complete Media and added to each well with A375 tumor cells. Prior to the addition of the T cells, the tandem BsAbs were titrated into the plates with tumor cells starting at 500 nM and ending at 10 pM. The T cell/tumor cell/BsAb (or MAb) mixtures were allowed to incubate for 24 h, then the plates were washed gently (twice) with serum free RPMI 1640. Finally, to determine the amount of tumor cells alive at the end of the incubations, 100 µL RPMI 1640 and 100 µL Cell Titer Glo reagent (Promega) were added, mixed for 2 min on a shaker (slowly) and incubated for 10 min in the dark. Lastly, the luminescence was read on an EnVision 2130 multilabel reader.

**Flow cytometry**. Cell culture was performed as described above. The day before running flow cytometry, T25 flasks were seeded with Saos-2 or 624.38 cells in Growth buffer (RPMI 1640 /10% fetal bovine serum (FBS) Corning/gentamicin—Gibco). The next morning, the Saos-2 and 624.38 tumor cells were washed once with Growth buffer. Peptide was added to the tumor cells (or not as a negative control) at 6 µg/mL for 3 h at 37 °C, 5% $CO_2$. Next, excess peptide was aspirated off and the cells were washed 3X with PBS buffer. The tumor cells were lifted from the T25 flasks using Accutase (Innovative Technologies Cat#AT104). All subsequent steps were performed on ice. The Jurkat cells grow in suspension (ATCC Cat#TIB-152). Cells were transferred to centrifuge tubes and pelleted by centrifugation at 170$g$ for 7 min. Henceforth, "Wash buffer" was PBS/2% FBS/0.05% $NaN_3$ /10% Normal Goat Serum with extra 10% FBS. "Blocking buffer" was Wash buffer supplemented with Human BD Fc Block (BD Biosciences Cat#564220). The cells were resuspended in Blocking buffer for 15 min, pelleted, Washed 3×, and resuspended in Wash buffer before adding 50 µL of the cells ($0.2 \times 10^6$ cells/well for tumor cells and $0.5 \times 10^6$ cells/well for Jurkat cells) to 96-well plates (Corning 3799). For the tumor cells, the TCR/CD3 bifunctionals and an anti-CD3 mAb (in-house, recombinantly-produced anti-CD3 chimeric SP34 hIgG1_N297Q[38]) were added to the wells at 30, 3, 0.3, and 0.03 µg/mL and incubated 45 min. For Jurkat cells, the TCR/CD3 bifunctional was titrated starting at 30 µg/mL using 3-fold dilutions. The cells were pelleted, washed, and the supernatants were aspirated again before adding 100 µL R-Phycoerythrin-conjugated Goat anti-Human Lambda (1:500 dilution, Southern Biotechnology Cat#2070-09, lot#B4419-P219) in Wash buffer for 45 min. The cells were pelleted and washed again. Finally, the cells were resuspended in Wash buffer with 1:1000 PI (Molecular Probes Cat#P3566) and covered with foil. The cells were then acquired on a Becton Dickinson LSRFortessa flow cytometer with BD FACSDiva software (v8.0.1) and data were analyzed using FlowJo version 10.5.3. The gating strategies for distinguishing live versus dead cells and single cells versus doublets/multiplets that were used for Jurkat, 624.38, and Saos-2 cells are provided in Supplementary Fig. S5-S7, respectively.

**Reporting summary**. Further information on research design is available in the Nature Research Reporting Summary linked to this article.

## Data availability
Source data are available in the Source Data file. The coordinates for the TCR constant domain structure, and the corresponding structure factors, have been deposited in the Protein Data Bank (http://www.rcsb.org) under accession code 6U07 [http://www.rcsb.org/structure/6U07]

All other relevant data are available upon reasonable request.

## Code availability
Custom Scripts for running computational single residue mutations in Rosetta are provided in the Supplementary Information as well as in the Rosetta Commons: rosetta_scripts_scripts/scripts/public/point_mutant_scan/point_mutant_scan. Froning_et_al.xml. To run the scripts, users must have access to Rosetta software [https://www.rosettacommons.org/software/license-and-download]. The scripts can also be found at [https://gist.github.com/JackMaguire/6f33119ff9cc4e46e16dc860cdf306e2]

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

## Acknowledgements

The authors would like to thank Shyama Yallapragada for her help with DNA construction and preparation, Nichole Niemela and Bon Gil Koo for CHO cell transfections, culture, and small scale purification, Michael Bacica for initial LCMS characterization, Carina Torres for flow cytometry discussions, and Andrew Leaver-Fay for helpful discussions. This research used resources of the Advanced Photon Source (APS), a U.S. Department of Energy (DOE) Office of Science User Facility operated for the DOE Office of Science by Argonne National Laboratory under Contract No. DE-AC02-06CH11357. We thank Jordi Benach, John Koss, and Laura Morisco for data collection support at the Lilly Research Laboratories Collaborative Access Team (LRL-CAT), beamline Sector 31, of the APS.

## Author contributions

K.F., J.M., B.K., and S.J.D conceived the project. K.F. and S.J.D. designed the molecular constructs. K.F. and J.D. performed the molecular biology, expression screening and K.F. and S.J.D. influenced downstream assays. J.M. and B.K. did the molecular modeling. S.C. designed and performed the DSF experiments. F.H. purified the proteins and performed the SDS-PAGE, analytical SEC, and DSC experiments. F.H. also deglycosylated and purified the Cα/Cβ constructs for crystallography. X.W. designed and constructed some of the original TCR constructs. A.J.F. collected and processed the X-ray data. K.W. conducted all crystallization experiments, including crystal seeding and harvesting for data collection. M.T.H. was responsible for crystal structure determination, refinement, and validation. J.R.F. designed, performed, and analyzed the peptide mass mapping experiments. D.B. designed, performed, and analyzed the HDX experiments. A.S. designed and performed the flow cytometry and redirected lysis assays. K.F., J.M, B.K., and S.J.D. wrote the manuscript except for some of the methods, which were provided by the other co-authors.

## Competing Interests

K.F., A.S., F.H., S.C., K.W., A.F., X.W., H.A., E.C., J.F., A.R.H., D.B., M.H., and S.J.D. are employees of Eli Lilly and Company. Eli Lilly and Company and the University of North Carolina at Chapel Hill filed a joint patent application (still unpublished) related to the TCR constant domain mutations and S.J.D., B.K., J.M., and K.F. are listed as inventors.
