## [Peer Review File · Nature Communications]

Reviewers' Comments:

Reviewer #1:

Remarks to the Author:

Froning et al. targeted the constant domain of TCRs for stability design. Unlike antibodies, which can often be produced in large quantities and are stable, TCRs are typically quite difficult in both regards. Using a combination of evolutionary-conservation analysis and Rosetta atomistic design, they selected dozens of point mutations for experimental testing, then based on the feedback from single-mutation experimental analysis, generated a mutant with seven mutations at the alpha/beta interface and then went on to demonstrate substantial thermal stabilisation (20oC) and an improvement in expressibility (approx. fivefold). They then showed that these mutations are transferrable to other TCRs and are therefore likely to be universal and also demonstrated how such a stabilised TCR could be combined with an antibody to generate a bispecific. This is an interesting application of computational design to an important protein-engineering problem.

Major comments:

* I had a hard time understanding the bottom lines of the immunology work on the bispecifics. The section dealing with this aspect of the work is written somewhat vaguely (especially the last paragraph of the Results) and without providing the bottom lines. For instance, the abstract says that the TCR/CD3 bispecifics can redirect T cells to kill tumor cells expressing a specific pMHC antigen. Was this actually observed experimentally or is this part of the vision for the future? If the latter, I think the authors need to drop the claim in the abstract. At any rate, they should make the bottom lines of this section clearer.

* The stabilised bsAb is present in the blood for seven days, then drops precipitously with an increase in anti-drug antibodies. Can the authors contextualise whether seven days would be sufficient in a realistic setting? Is the ADA worrying?

* The authors noted a significant drop in affinity in the bsAb due to the lack of avidity. Is the resulting affinity realistic for practical use?

* In the introduction, the authors point out that they tested both evolutionary-constrained and "pure" Rosetta design and combined mutations from both approaches. This could lead to interesting lessons, but in the Results, the authors go over the seven mutations that they introduced without drawing clear lessons about what works and what fails. From my reading, it appears that the two hydrophobic mutations were very successful and accurately modelled and did not come from the evolutionary analysis; by contrast, while the polar mutations were somewhat stabilising, the crystal structure revealed substantial errors in modelling. If this is indeed the case, I recommend providing this as a (tentative) bottom line of the design aspect of this work: atomistic design calculations work well with hydrophobic mutations but an MSA is required to make polar mutations and that even then, some inaccuracy is to be expected. This means that the design should be restricted to areas that are not directly involved in binding. Also, the authors provide no information about the majority of the point mutations that were tested experimentally and which led to no stabilisation. Are there commonalities among these mutations?

Minor comments:

* Abstract: "...perhaps inhibited by conformational rigidity". I think I understand what they mean but it's not clear what is the subject of "inhibited".

* line 47: BiTE-like bispecifics. The acronym was not previously introduced.

* It may be difficult, but I wish that the authors provided a little bit more information on the subjective rules they used in selecting sequences to go into the alignment and in choosing the mutants for experimental testing. Even if the rules are not quantitative, some information may be useful to others.

* line 175: Mutations to proline. May be good to cite Brian Matthews's work on stabilisation through Pro mutations.

- * line 270: 90% of the stabilised bispecific was secreted in pure form after affinity chromatography. How does this compare to standard formats of bispecific antibodies? Is this useful in a translational setting?
- * line 286: The authors state the bsAb antibody's halflife with a very large margin of error (3-8 days) "depending on the timepoints included in the fits". If the fits are so sensitive, maybe the data are not reliable? Should this experiment be redone or dropped from the paper?
- * Fig2, where the structure and design models are compared is unclear. Since the backbone movements are quite small between the design and structure, the authors could concentrate only on the rotamer changes. It's quite difficult to make out where the mutations occur with respect to the alpha/beta interface. They should also provide more detail on neighbouring residues for these mutations. Last, since most of the inaccurate mutations are polar, I wonder if water molecules are playing an important role in explaining the discrepancies between the model and structure. If so, this should be indicated in the figure and in the text.
- * The provided Rosetta script is only for all-atom minimisation without coordinate constraints. Could the authors provide all the other scripts and commandlines?

Reviewer #2:

Remarks to the Author:

In this study the authors conduct an in-depth analysis of rationally engineered TCR constant domains to increase the stability and expression of 4 TCRs. They conducted an in depth biophysical analysis of the designed molecules, including their post-translational modifications and demonstrated their efficacy. I felt this was a very well designed and implemented study, and look forward to its publication. I only have a few minor comments.

1. The production of bispecific molecules can often times be challenging in part due to limitations of stability. They solved this issue, but their claim of "antibody-like expression" is vague and overstated. Their protein titers were very low compared to industrial antibodies (averaging more than 3-5 g/L in industrial production from CHO cells, when produced in fed-batch), which their best titer was an order of magnitude lower. I'd recommend just removing the claim. However, I think the gain they showed in titer is of substantial value given the difficulty of producing engineered bispecific molecules.
2. Line 231 "structured" should be "structure"
3. Line 267 "mutaions" should be "mutations"

Reviewer #3:

Remarks to the Author:

Recombinant T cell receptors (TCRs) have therapeutic potential by redirecting naïve T cells to recognize and eliminate non-self antigen presenting cells (like virally infected or cancerous cells). Production of TCRs is hampered however by low stability and unreliable expression levels. In this manuscript, Froning and colleagues aim to find point mutations that would be a generic stabilizing approach for a/b TCRs.

They use molecular modeling software Rosetta to identify seven point mutations in the TCR constant domains (Ca/Cb) that together (and combined with a previously reported di-sulfide bond Ca166/Cb173) significantly improve expression and stabilize the CaCb subunits (as judged by Differential scanning calorimetry). The mutated CaCb subunits also had a smaller apparent size and were less glycosylated, which seems to suggest a more compact and rigid structure. When incorporated in unrelated full length TCRs there was a modest, but reproducible increase in stability after the addition of the 7 identified mutations, which now also allowed for deleterious

mutations in the variable regions.

The authors solve the crystal structure of the mutated CaCb and comparison with the published wild-type structure showed (as expected) good agreement with predictions by the Rosetta program for the hydrophobic mutations, but not the hydrophilic ones (which they speculate is due to too stringently restricting backbone movement).

They generated both IgG-like and Tandem Fab-like bispecific antibodies, and the authors could show that both are potent in binding CD3 and HLA-A2/peptide presenting cells.

Overall, I find the study generally well executed and presented. This work could pave the way for the production of (novel) TCRs (or derivations thereof) and could therefore find an audience in both academia and industry.

Minor points:

- Please explain what was used to determine the increase of expression (for example in line 139, but many more) and show data.
- Fig 1: There is no reference to the orange curve in the figure legend
- The error bars in fig 1b are missing
- As a non-user of the commercial EpiVax server, the paragraph starting at line 147 is hard to understand. Perhaps the authors can consider explaining term as for example 'EpiMatrix score', 'Epimatrix cluster' or 'Epibars'. Also, statements as 'significant changes' and 'subtle increases' are hard to interpret without any data or reference.
- Add CC1/2 to crystallographic table
- Line 197, please add rmsd when stating backbone conformations of two structures are 'similar'
- Fig 3 and 4, add labels to additional residues you are referring to
- Fig 6c, what does 'polished' mean?
- Table 1: some cells display multiple numbers, what are they?

Reviewer #4:

Remarks to the Author:

In this study the authors explored the possibility to mutate the TCR α and β constant region sequences to improve the yield of producing recombinant TCR molecules with enhanced stability. They combined the use of Rosetta software and multiple sequence alignment to identify candidate residues. The production of recombinant protein showed that a combination of 7 amino-acid residues, 3 in the constant α domain and 4 in the β domain, improved the yield and stability of recombinant TCR molecules.

The authors have performed extensive biochemical characterization of wild type TCR molecules and molecules with the change of the identified 7 amino-acid residues. While the biochemical dataset is extensive, the functional dataset is poor.

Points to consider:

- 1) The α T166C/ β S173C modification has been used to date extensively to improve soluble TCR production and TCR expression in gene engineered T cells. The authors should compare how the new Des mutations compare with the α T166C/ β S173C mutations. There is only one example in table 1 where this has been compared. In the wild type NY-ESO-1 TCR the introduction of Des boosts yield more than the introduction α T166C/ β S173C. However, both changes improve protein stability to similar levels. It is important to understand whether the Des modification primarily improves yield, while the effect on stability is similar to that seen with α T166C/ β S173C.
- 2) Most Des modifications were done in "WT-TCR" that already have the α T166C/ β S173C modifications. They are in fact not wild type TCRs. It is therefore not possible to judge how much Des improves the performance of wild type TCRs.

- 3) Figure 5 d shows that Des improves the yield of all TCRs tested, but panel e) indicates that in some TCRs the stability is decreased. Reduced stability of soluble TCRs would be a substantial concern in the therapeutic setting; this has not been considered by the authors.
- 4) The PK studies in figure S2 should include molecules that do not have the Des modifications. Both molecules tested carry the modifications and it is therefore not possible to judge whether Des gives an improvement.
- 5) The staining experiments in figure 6d should include a tumor cell that expresses NY-ESO-1 endogenously to test whether natural levels of expression result in staining.
- 6) The redirected killing experiments in figure 6e should include a tumor cell that expresses HLA-A*0201 plus NY-ESO-1 and HLA-A*0201 without NY-ESO-1. This will determine whether there is antigen-specific killing of cells expressing NY-ESO-1.

Reviewers' comments:

Reviewer #1 (Remarks to the Author):

Froning et al. targeted the constant domain of TCRs for stability design. Unlike antibodies, which can often be produced in large quantities and are stable, TCRs are typically quite difficult in both regards. Using a combination of evolutionary-conservation analysis and Rosetta atomistic design, they selected dozens of point mutations for experimental testing, then based on the feedback from single-mutation experimental analysis, generated a mutant with seven mutations at the alpha/beta interface and then went on to demonstrate substantial thermal stabilisation (20oC) and an improvement in expressibility (approx. fivefold). They then showed that these mutations are transferrable to other TCRs and are therefore likely to be universal and also demonstrated how such a stabilised TCR could be combined with an antibody to generate a bispecific. This is an interesting application of computational design to an important protein-engineering problem.

Major comments:

** I had a hard time understanding the bottom lines of the immunology work on the bispecifics. The section dealing with this aspect of the work is written somewhat vaguely (especially the last paragraph of the Results) and without providing the bottom lines. For instance, the abstract says that the TCR/CD3 bispecifics can redirect T cells to kill tumor cells expressing a specific pMHC antigen. Was this actually observed experimentally or is this part of the vision for the future? If the latter, I think the authors need to drop the claim in the abstract. At any rate, they should make the bottom lines of this section clearer.*

Response: Data supporting the biological claims was provided in the original **Figure 6** (updated now to Fig 7) including tumor cell and T cell binding as well as TCR/CD3 Bispecific-mediated redirection of T cells to kill tumor cells. Specifically, peptide/MHC-binding to two HLA-A2+ cell lines was shown in the original **Figure 6d** and TCR/CD3 bispecific-mediated redirecting of T cells to kill HLA-A2+ Saos-2 sarcoma cells by was shown in the original **Figure 6e**. However, the lack of details is a good comment. Therefore, we included flow cytometry binding titrations of the TCR/CD3 bispecifics in the absence and presence of exogenous NY-ESO-1 peptide with the tumor cells to demonstrate specific binding to the MHC/peptide complex (**New Figure 7a**). We also provided experiments evaluating the ability of the TCR/CD3 bispecifics to redirect T cells to kill both Saos-2 and 624.38 HLA-A2+ tumor cell lines in the absence and presence of exogenous NY-ESO-1 peptide (**New Figure 7b**). We significantly expanded the functional cell biology section to include more details. We provide detailed gating strategies used for the flow cytometry experiments in 3 additional **Supplementary Figures (Fig. S5-S7)**. Lastly, we provide a supplementary figure, **Fig. S4**, demonstrating the impact of increasing the affinity of the TCR on the redirected lysis activity of the IgG format of the TCR/CD3 bispecific.

** The stabilised bsAb is present in the blood for seven days, then drops precipitously with an increase in anti-drug antibodies. Can the authors contextualise whether seven days would be sufficient in a realistic setting? Is the ADA worrying?*

Response: Mouse ADA developed towards human proteins in mice is common. Observing ADA at 7 days is less common, but we have witnessed it with other human biologics. Often there is a mechanism for internalization on antigen presenting cells that is required, but we have not looked into this. However, we show later in the manuscript that the IgG bispecific format showed inferior T cell redirected lysis of tumor cells, which significantly reduces its future utility. Therefore, the improved pharmacokinetic properties of the IgG form were nullified by its poor activity. For this reason, we decided to remove the pharmacokinetic experiments from the manuscript.

** The authors noted a significant drop in affinity in the bsAb due to the lack of avidity. Is the resulting affinity realistic for practical use?*

Response: The drop in EC50 binding by flow cytometry due to a switch from 2 arm binding to 1 arm binding was expected. CD3 affinity need not be high to enable redirected lysis. We demonstrate in the original **Fig. 6** (updated now to **Fig 7**) that the CD3 arm is still capable of redirecting T cells to kill tumor cells, thus demonstrating that one-arm binding is realistic for practical use. In fact, high affinity CD3 binding can promote wide-spread T cell activation in the absence of antigen, systemic cytokine release, and toxicity; thus, low affinity 1 arm binding to CD3 is preferred for T cell redirection. Blinatumomab, the only approved Tumor associated antigen (CD19)/CD3 bispecific has a low affinity for CD3 => 100 nM (PMID:23623807).

** In the introduction, the authors point out that they tested both evolutionary-constrained and "pure" Rosetta design and combined mutations from both approaches. This could lead to interesting lessons, but in the Results, the authors go over the seven mutations that they introduced without drawing clear lessons about what works and what fails. From my reading, it appears that the two hydrophobic mutations were very successful and accurately modelled and did not come from the evolutionary analysis; by contrast, while the polar mutations were somewhat stabilising, the crystal structure revealed*

substantial errors in modelling. If this is indeed the case, I recommend providing this as a (tentative) bottom line of the design aspect of this work: atomistic design calculations work well with hydrophobic mutations but an MSA is required to make polar mutations and that even then, some inaccuracy is to be expected. This means that the design should be restricted to areas that are not directly involved in binding.

Response: We agree with the reviewer's assessment that modeling polar amino acids is more challenging and therefore MSA information may be more useful in these cases. We have added the following text to the end of the results section that discusses the mutations:

"Overall, the stability measurements and the crystal structure of the stabilized variant highlight some of the strengths and weaknesses of protein modeling. In general, it is more straightforward to predict the conformation and energetic consequences of hydrophobic mutations in the protein core. The constrained environment as well as the more limited flexibility of hydrophobic side chains allows more accurate prediction of the most favorable rotamer. Polar amino acids are typically exposed to solvent and in general are longer with more sidechain degrees of freedom. Given the challenges associated with modeling polar residues, making use of information in a MSA may be most useful when mutating polar amino acids. Consistent with this conclusion, all five of the stabilizing mutations that involved mutation to a polar amino acid were present in the MSA, while the stabilizing mutations that were not in the MSA involved mutations to hydrophobic amino acids. The absolute value of the Rosetta Energies (dE) for each of the stabilizing polar mutations was smaller than all the dEs of the mutations chosen without the requirement of being within the MSA. Without the MSA filter, most of the stabilizing polar mutations would have been included in the group of 50 mutants that were screened."

** Also, the authors provide no information about the majority of the point mutations that were tested experimentally and which led to no stabilisation. Are there commonalities among these mutations?*

Response: In **Table S1** we list the Tms for the 16 point mutations that were characterized with thermal unfolding experiments. 7 mutations increase the Tm by more than 2 degrees. The remaining 9 mutations have Tms within 1 degree of the wild type protein. We are hesitant to draw broad conclusions from a data set of this size, but overall we were excited that most of the mutations either boosted stability or were neutral. There were also a number of mutations for which we did not perform thermal unfolding experiments because they did not boost protein expression. Experience with other systems suggests that this does not mean that these proteins were destabilized, as protein expression can depend on many factors.

Minor comments:

* Abstract: "...perhaps inhibited by conformational rigidity". I think I understand what they mean but it's not clear what is the subject of "inhibited".

Response: We modified this sentence of the abstract to be more specific: '...perhaps through conformational stabilizations that restrict access to N-linked glycosylation enzymes.'

* line 47: BiTE-like bispecifics. The acronym was not previously introduced.

Response: We defined the acronym BiTE (Bispecific T cell engager) and modified the sentence appropriately.

** It may be difficult, but I wish that the authors provided a little bit more information on the subjective rules they used in selecting sequences to go into the alignment and in choosing the mutants for experimental testing. Even if the rules are not quantitative, some information may be useful to others.*

Response: As the reviewer implies, there was some subjectivity in which sequences were included in our TCR sequence alignment. We only allowed sequences from animals and we removed sequences if the alignment was poor or because the sequence was labeled as something other than a TCR. We now mention these criteria in the methods section.

To choose mutations for experimental testing we made use of a plot that we now include in the supplementary material and reference in the methods section (**NEW Fig. S1**). This figure shows two computational metrics for all possible single point mutants in the constant domains: the calculated Rosetta score for the mutant and a log odds score based on the mutation's representation in the MSA. As can be observed, we primarily chose mutations based on Rosetta score but included some mutations because they had better log odds scores and were still predicted to be favorable according to Rosetta. The plot also illustrates that some of the experimentally favorable mutations that were identified were present in the MSA while others were not.

* line 175: Mutations to proline. May be good to cite Brian Matthews's work on stabilisation through Pro mutations.

Response: We now include this citation.

* line 270: 90% of the stabilised bispecific was secreted in pure form after affinity chromatography. How does this compare to standard formats of bispecific antibodies? Is this useful in a translational setting?

Response: Great question. We added two sentences in the TCR/CD3 purification Results section describing the clinical and commercial translation impact.

'The purity of the stabilized TCR/CD3 BsAbs directly off an affinity chromatography step is remarkable, even for antibody-based bispecifics (PMID:18729019). Both the improved expression and optimal purity would have a big impact on translatability to the clinic and for commercializability of the platform since the cost of biologics manufacturing is high and the removal of product-related impurities can be challenging and result in significantly reduced overall product yields.'

* line 286: The authors state the bsAb antibody's half-life with a very large margin of error (3-8 days) "depending on the timepoints included in the fits". If the fits are so sensitive, maybe the data are not reliable? Should this experiment be redone or dropped from the paper?

Response: As described above, we removed the pharmacokinetic experiment given that the TCR/CD3 IgG BsAb had poor T cell redirected lysis activity (Old figure 6 – New Figure 7) rendering the pharmacokinetic properties of the IgG BsAb less interesting.

** Fig2, where the structure and design models are compared is unclear. Since the backbone movements are quite small between the design and structure, the authors could concentrate only on the rotamer changes. It's quite difficult to make out where the mutations occur with respect to the alpha/beta interface. They should also provide more detail on neighbouring residues for these mutations. Last, since most of the inaccurate mutations are polar, I wonder if water molecules are playing an important role in explaining the discrepancies between the model and structure. If so, this should be indicated in the figure and in the text.*

Response: Thank you for the suggestions. We have made several changes to **Fig. 2** and **Fig. 3** to better illustrate the tertiary environment of the mutations. First, we include an overview panel that shows the location of the mutations in the context of the full-length TCR bound to a peptide/MHC complex. For the snapshots of each mutation we now show more neighboring residues in line representation, label key interacting residues, and display hydrogen bonds to waters that are present in the structures.

** The provided Rosetta script is only for all-atom minimisation without coordinate constraints. Could the authors provide all the other scripts and commandlines?*

Response: We now provide the Rosetta script that was used to evaluate each mutation (including coordinate constraints that were used), a sample resfile used to control which neighboring residues could move, a portion of code used to generate the resfile, and an example command line.

Reviewer #2 (Remarks to the Author):

In this study the authors conduct an in-depth analysis of rationally engineered TCR constant domains to increase the stability and expression of 4 TCRs. They conducted an in depth biophysical analysis of the designed molecules, including their post-translational modifications and demonstrated their efficacy. I felt this was a very well designed and implemented study, and look forward to its publication. I only have a few minor comments.

1. The production of bispecific molecules can often times be challenging in part due to limitations of stability. They solved this issue, but their claim of "antibody-like expression" is vague and overstated. Their protein titers were very low compared to industrial antibodies (averaging more than 3-5 g/L in industrial production from CHO cells, when produced in fed-batch), which their best titer was an order of magnitude lower. I'd recommend just removing the claim. However, I think the gain they showed in titer is of substantial value given the difficulty of producing engineered bispecific molecules.

Response: Good comment. High g/L titers are only typical for stable cell mammalian lines after weeks of selection and screening using various strategies that select for cells with optimal gene integration. The titers we report are for *transient* transfection in CHO cells, which are typically much lower in our hands. Protein titers can vary depending on the health of the cells, subtle differences in media, and other

factors. But to provide a more concrete expression comparison with monoclonal antibodies (mAbs), we performed additional experiments.

We expressed 3 non-stabilized and 3 stabilized TCRs (all lacking the $\alpha 166/\beta 173$ disulfide) simultaneously alongside 7 different human monoclonal antibodies (mAbs). The average mAb titer was 235 ± 280 mg/L (one mAb expressed at ~ 800 mg/L, while the others expressed between 100-400 mg/L, which explains the high error). The average non-stabilized TCR expression level without the designs was 59 ± 32 mg/L while, the average TCR expression with the stability designs was 284 ± 176 mg/L. Additionally, the quality of the TCRs lacking the stability designs was clearly much poorer than those with the designs when assessed by SDS-PAGE and differential scanning fluorimetry. This new data was added to **Table 1**, the results were added to the '**Impact of the $C\alpha/C\beta$ Designs on Full Length TCRs**' Results section, and a new Supplementary figure was added that provides the characterization of these TCRs lacking the $\alpha 166/\beta 173$ disulfide (Now **Fig. S2**).

2. Line 231 "structured" should be "structure"

Response: Fixed

3. Line 267 "mutaions" should be "mutations"

Response: Fixed

Reviewer #3 (Remarks to the Author):

Recombinant T cell receptors (TCRs) have therapeutic potential by redirecting naïve T cells to recognize and eliminate non-self antigen presenting cells (like virally infected or cancerous cells). Production of TCRs is hampered however by low stability and unreliable expression levels. In this manuscript, Froning and colleagues aim to find point mutations that would be a generic stabilizing approach for a/b TCRs.

They use molecular modeling software Rosetta to identify seven point mutations in the TCR constant domains (Ca/Cb) that together (and combined with a previously reported di-sulfide bond Ca166/Cb173) significantly improve expression and stabilize the CaCb subunits (as judged by Differential scanning calorimetry). The mutated CaCb subunits also had a smaller apparent size and were less glycosylated, which seems to suggest a more compact and rigid structure. When incorporated in unrelated full length TCRs there was a modest, but reproducible increase in stability after the addition of the 7 identified mutations, which now also allowed for deleterious mutations in the variable regions.

The authors solve the crystal structure of the mutated CaCb and comparison with the published wild-type structure showed (as expected) good agreement with predictions by the Rosetta program for the hydrophobic mutations, but not the hydrophilic ones (which they speculate is due to too stringently restricting backbone movement).

They generated both IgG-like and Tandem Fab-like bispecific antibodies, and the authors could show that both are potent in binding CD3 and HLA-A2/peptide presenting cells.

Overall, I find the study generally well executed and presented. This work could pave the way for the

production of (novel) TCRs (or derivations thereof) and could therefore find an audience in both academia and industry.

Minor points:

- Please explain what was used to determine the increase of expression (for example in line 139, but many more) and show data.

Response: Expression titers were assessed using an Octet Red instrument with Ni²⁺-NTA sorters and purified C α /C β or full-length α/β TCR with C-terminal C α -histags as controls. This was described in one sentence in the methods, but we added the following to the results when first describing expression levels: 'Expression was assessed using an Octet Red with Ni²⁺-NTA sorters and purified C α /C β protein containing a C-terminal C α -histag.'

- Fig 1: There is no reference to the orange curve in the figure legend.

Response: The orange curve is referenced in the figure legend as the C α /C β constant domain subunit.

- The error bars in fig 1b are missing.

Response: Except for the initial wild-type control, the point mutants were only expressed at scale, purified, and tested once as part of the initial screening therefore, we do not have multiple experiments to provide error bars. The typical error of these experiments is roughly 1 °C. We added information to the figure legend.

- As a non-user of the commercial EpiVax server, the paragraph starting at line 147 is hard to understand. Perhaps the authors can consider explaining term as for example 'EpiMatrix score', 'Epimatrix cluster' or 'Epibars'. Also, statements as 'significant changes' and 'subtle increases' are hard to interpret without any data or reference.

Response: We re-worded the EpiVax paragraph to provide interpretation of the EpiVax scores – and simplified the paragraph. The EpiVax reference is provided.

'Lastly, we assessed the impact of the designs on potential immunogenicity of the TCR. Using the EpiVax server, we investigated whether any increase MHC-peptide complexes were likely for each of the mutations [reference 23 - PMID:19269256]. Overall, a slight increase in the EpiMatrix score was observed for both C α and C β . Increases in the individual peptide scores were marginal and did not reach the level predicted to result in significant MHC binding. An outlier of concern included the C α _T150I mutation, whose score went up substantially for one peptide 9-mer and there is strong predicted MHC binding predicted for a second 9-mer peptide that also exists in the wild-type C α peptide. The C β _E134K_H139R mutants are found together on multiple 9-mer peptides and there is a slight increase in the EpiMatrix scores, but none of the scores put these peptides in the high-risk category (i.e., strong predicted MHC binding). Interestingly, the C β _D155P and C β _S170D variants, which are the two most impactful mutants from a T_m perspective, lower the overall EpiMatrix scores versus wild-type C α /C β .'

- Add CC1/2 to crystallographic table –

Response: Done

- Line 197, please add rmsd when stating backbone conformations of two structures are 'similar'

Response: We now provide the rmsd: 0.338 angstroms.

- Fig 3 and 4, add labels to additional residues you are referring to –

Response: As noted in our response to reviewer 1, we made several changes to these figures in order to better illustrate the environment of the mutations. These changes included labels on key neighboring residues.

- Fig 6c, what does 'polished' mean?

Response: This means purified through a second method/resin. We changed the language to be more explicit.

- Table 1: some cells display multiple numbers, what are they?

Response: They are duplicates. To simplify, we averaged the values and added the difference between the two values in Table 1.

Reviewer #4 (Remarks to the Author):

In this study the authors explored the possibility to mutate the TCR α and β constant region sequences to improve the yield of producing recombinant TCR molecules with enhanced stability. They combined the use of Rosetta software and multiple sequence alignment to identify candidate residues. The production of recombinant protein showed that a combination of 7 amino-acid residues, 3 in the constant α domain and 4 in the β domain, improved the yield and stability of recombinant TCR molecules.

The authors have performed extensive biochemical characterization of wild type TCR molecules and molecules with the change of the identified 7 amino-acid residues. While the biochemical dataset is extensive, the functional dataset is poor.

Points to consider:

1) The α T166C/ β S173C modification has been used to date extensively to improve soluble TCR production and TCR expression in gene engineered T cells. The authors should compare how the new Des mutations compare with the α T166C/ β S173C mutations. There is only one example in table 1 where this has been compared. In the wild type NY-ESO-1 TCR the introduction of Des boosts yield more than the introduction α T166C/ β S173C. However, both changes improve protein stability to similar levels. It is important to understand whether the Des modification primarily improves yield, while the effect on stability is similar to that seen with α T166C/ β S173C.

Response: Great suggestion. In response, we made the three other TCRs with and without the 7 mutant stabilizing designs, **but without the 166/173 disulfide**. We performed additional expressions and characterized the proteins. The data was added to **Table 1** and the characterization was provided in a **new Supplementary Figure S2**. The seven mutant designs significantly increase TCR expression in the absence of the 166/173 disulfide. Except for the NY-ESO-1 TCR, where the 7 mutant designs led to roughly the same stability increase as the 166/173 disulfide, the 7 mutant designs led to higher increases in TCR stability than the 166/173 disulfide. We folded these results into the **'Impact of the C α /C β Designs on Full Length TCRs'** section.

2) Most Des modifications were done in "WT-TCR" that already have the α T166C/ β S173C modifications. They are in fact not wild type TCRs. It is therefore not possible to judge how much Des improves the performance of wild type TCRs.

Response: This comment was also addressed in response to the previous comment. Please see above.

3) Figure 5 d shows that Des improves the yield of all TCRs tested, but panel e) indicates that in some TCRs the stability is decreased. Reduced stability of soluble TCRs would be a substantial concern in the therapeutic setting; this has not been considered by the authors.

Response: Good question. With the less stable constant domains – it is difficult to resolve the stability of the variable versus the constant domains – whereas it is easy to pick out with the stabilized constant domains. We believe the subtle difference (~ 1 °C) in the measured T_ms is the result of our inability to resolve the T_m of the variable domains in the non-stabilized TCRs, which is close to that of the constant domains. We do not believe that the designs are reducing the stability of the TCRs in any instance. We added the following sentence to the Results paragraph describing Figure 5:

'A slightly higher apparent T_m of the non-stabilized anti-NY-ESO-1 TCR containing β N66E versus that of the TCR containing the 7 mutant designs is unlikely a real difference, but instead the result of the overlap of the unfolding transitions of the variable and constant domains within the non-stabilized TCR making it difficult to accurately measure the variable domain T_m.'

4) The PK studies in figure S2 should include molecules that do not have the Des modifications. Both molecules tested carry the modifications and it is therefore not possible to judge whether Des gives an improvement.

Response: Given that the activity of the IgG format of the TCR/CD3 BsAbs was significantly inferior to that of the tandem Fab format (roughly 100-1000-fold), we decided the discussion of the pharmacokinetics of an inferior format provided little value. Therefore, we removed the pharmacokinetic studies.

5) The staining experiments in figure 6d should include a tumor cell that expresses NY-ESO-1 endogenously to test whether natural levels of expression result in staining.

Response: Both the 624.38 and Saos-2 endogenously express NY-ESO-1 antigen and HLA-A2 (reference provided in the TCR/CD3 bifunctional section - PMID:15778407). However, the cleaved peptide does not reach the cell surface to an extent that allows for either flow cytometric staining or T cell redirected lysis. We added our flow cytometry data demonstrating the low intrinsic binding of the TCR/CD3 bispecifics to the tumor cells in the absence of pulsed exogenous peptide and also redid the redirected lysis experiments in the presence AND absence of the pulsed peptide. We add a literature reference indicating that pulsing the peptide in vitro is often required to observe redirected lysis even for primary T cells expressing recombinant α/β TCRs. To make an actual therapeutic, it would be necessary to find a highly potent TCR that can kill tumor cells displaying endogenous levels of peptide/MHC complexes, but this is outside the scope of this study.

*6) The redirected killing experiments in figure 6e should include a tumor cell that expresses HLA-A*0201 plus NY-ESO-1 and HLA-A*0201 without NY-ESO-1. This will determine whether there is antigen-specific killing of cells expressing NY-ESO-1.*

Response: The new data provided for comment 5) showed that no redirected lysis occurs in the absence of pulsed NY-ESO-1 peptide *even though* the two cell lines chosen for the analysis are HLA-A2+/NY-ESO-1+. The goal of this manuscript was not to prove the specificity of this particular TCR, whose specificity and binding have been characterized extensively in the literature, but rather to demonstrate the improved biophysical properties that lead to improved production of these bifunctionals in mammalian cells.

Reviewers' Comments:

Reviewer #1:

Remarks to the Author:

The authors addressed all of the points adequately. This is a very interesting contribution.

--

Sarel Fleishman

Reviewer #4:

Remarks to the Author:

The authors have addressed many points, but have not shown that their technology produces recombinant TCR/CD3 molecules that improve T cell killing of tumor cells or virus-infected cells. In the abstract the authors state "These TCR/CD3 bispecific proteins can redirect T cells to kill tumor cells with target HLA/peptide on their surfaces". This is misleading as it implies the killing of target cells that express the target protein. However, Fig 7b shows that TCR/CD3 molecules only work when cancer cells are pulsed with synthetic peptides. The expression of the target protein in the tumor cells is insufficient for the TCR/CD3 bispecific proteins to mediate T cell killing. In the rebuttal the authors state that "To make an actual therapeutic, it would be necessary to find a highly potent TCR that can kill tumor cells displaying endogenous levels of peptide/MHC complexes, but this is outside the scope of this study". This leaves the reader without any information whether the improved biochemical properties of the TCR/CD3 bispecific proteins translate into measureable improvement of redirected T cell killing of target cells.

03/11/20

Reviewer #4: Remarks to the Author: The authors have addressed many points, but have not shown that their technology produces recombinant TCR/CD3 molecules that improve T cell killing of tumor cells or virus-infected cells. In the abstract the authors state “These TCR/CD3 bispecific proteins can redirect T cells to kill tumor cells with target HLA/peptide on their surfaces”. This is misleading as it implies the killing of target cells that express the target protein. However, Fig 7b shows that TCR/CD3 molecules only work when cancer cells are pulsed with synthetic peptides. The expression of the target protein in the tumor cells is insufficient for the TCR/CD3 bispecific proteins to mediate T cell killing. In the rebuttal the authors state that “To make an actual therapeutic, it would be necessary to find a highly potent TCR that can kill tumor cells displaying endogenous levels of peptide/MHC complexes, but this is outside the scope of this study”. This leaves the reader without any information whether the improved biochemical properties of the TCR/CD3 bispecific proteins translate into measurable improvement of redirected T cell killing of target cells.

We felt this concern was *an assumption* that the bifunctionals could not have activity on tumor cell lines expressing endogenous levels of antigen. To address this concern, we performed experiments with a tumor cell line (A375) known to express the NY-ESO-1 antigen and demonstrated that the bifunctionals described in our study could redirect T cells to kill these tumor cells. This included the production and characterization of a new TCR/CD3 bifunctional molecule. We hope this dispels the assumption that the bifunctionals in our manuscript are not functionally active on tumor cells expressing endogenous antigen. These data are provided in **Figure 7c** and described in the final paragraph of the results.

Reviewers' Comments:

Reviewer #4:

Remarks to the Author:

The authors state in their reply that "Reviewer #4 expressed a new concern that the bifunctionals we generated could not kill tumor tumor cells without the addition of exogenous antigen (artificially) to assist in the redirection of T cells to kill the tumor cells"

I would like to point out that this is not a new concerns. Point 6 of the initial review asked the authors to address this point: "The redirected killing experiments in figure 6e should include a tumor cell that expresses HLA-A*0201 plus NY-ESO-1 and HLA-A*0201 without NY-ESO-1. This will determine whether there is antigen-specific killing of cells expressing NY-ESO-1." The authors now show killing of a tumor cell line that expresses HLA-A*0201 plus NY-ESO-1.

It would be useful if the authors could explain why a 40-fold affinity difference between the two tested TCR/CD3 bifunctional molecules does not result in any difference in the titration profiles in the killing assay shown in figure 7c.

04/02/20 Point by Point response to the Reviewers' comments:

Reviewer #4 (Remarks to the Author):

It would be useful if the authors could explain why a 40-fold affinity difference between the two tested TCR/CD3 bifunctional molecules does not result in any difference in the titration profiles in the killing assay shown in figure 7c.

Response: In the discussion, we add a sentence when comparing the geometrical constraints of TCR/CD3 BsAbs and their ability to mediate redirected lysis. We provide the hypothesis that once a certain TCR affinity is reached, the geometrical constraints of the TCR/CD3 bifunctional may become the limiting factor for potency.

“...enhancing the affinity of both the IgG and tandem Fab BsAb by 20- to 30-fold did not result in significantly improved potency, perhaps because the geometrical constraints of each format may dictate the maximum potency that can be observed once a certain affinity is reached.”